Subject Area:
biochemistry/cellular biology/molecular biology

Keywords:
spindle orientation, microtubules, actin

Author for correspondence:
Viji M. Draviam
e-mail: v.draviam@qmul.ac.uk

†Present address: Pengiran Anak Puteri Rashidah Sa'adatul Bolkiah Institute of Health Sciences, Universiti Brunei Darussalam, Jalan Tungku Link, Gadong BE 1410, Brunei Darussalam.
‡Present address: Center for Cancer Research, National Cancer Institute, National Institutes of Health, Bethesda, MD 20892, USA.

# MARK2/Par1b kinase present at centrosomes and retraction fibres corrects spindle off-centring induced by actin disassembly

Madeleine Hart[1], Ihsan Zulkipli[2,†], Roshan Lal Shrestha[2,‡], David Dang[1,3], Duccio Conti[1], Parveen Gul[1], Izabela Kujawiak[2] and Viji M. Draviam[1]

[1]School of Biological and Chemical Sciences, Queen Mary University of London, London, UK
[2]Department of Genetics, University of Cambridge, Cambridge, UK
[3]Department of Informatics, King's College, London, London, UK

DD, 0000-0001-8575-9021; DC, 0000-0003-4009-5940; VMD, 0000-0001-8295-3689

Tissue maintenance and development requires a directed plane of cell division. While it is clear that the division plane can be determined by retraction fibres that guide spindle movements, the precise molecular components of retraction fibres that control spindle movements remain unclear. We report MARK2/Par1b kinase as a novel component of actin-rich retraction fibres. A kinase-dead mutant of MARK2 reveals MARK2's ability to monitor subcellular actin status during interphase. During mitosis, MARK2's localization at actin-rich retraction fibres, but not the rest of the cortical membrane or centrosome, is dependent on its activity, highlighting a specialized spatial regulation of MARK2. By subtly perturbing the actin cytoskeleton, we reveal MARK2's role in correcting mitotic spindle off-centring induced by actin disassembly. We propose that MARK2 provides a molecular framework to integrate cortical signals and cytoskeletal changes in mitosis and interphase.

## 1. Introduction

Tissue development and regeneration requires a close coordination of cell division and cell polarity regulatory pathways. With key regulatory function in these pathways, PAR1/MARK is a family of kinases evolutionarily conserved from yeasts to humans (reviewed in [1,2]). Dysregulation of PAR1/MARK signalling has been implicated in a variety of human pathologies including neurodegenerative disorders, carcinomas and metabolic diseases, making it an attractive therapeutic target [1].

Among the four human Par1 kinases (MARK 1–4), MARK2/hPar1b is uniquely important for establishing the plane of division and it achieves this through two modes: a cell polarity-dependent mode and also a cell polarity-independent but cell shape-determined mode [3–5]. In both cases, MARK2 controls the position of the mitotic spindle, as seen in a variety of non-polarized and polarized systems, including human cervical epithelial cell cultures, hepatocyte lumina and columnar epithelia [3–6]. While cell polarity pathways control spindle positioning by asymmetric enrichment of cortical dynein, how interphase cell shape determines spindle positioning in non-polarized cultures is not fully clear (reviewed in [7,8]).

By imaging and modelling spindle movements in hundreds of unperturbed human epithelial cell cultures, we showed that the interphase cell shape can bias the rotation of mitotic spindles and, in turn, bias the plane of cell division [9,10]. Retraction fibres and adherens junctions retain the memory of interphase cell shape and tension [11–15]. Altering the distribution of retraction fibres using substrate micropatterning or laser surgery demonstrated the importance

of retraction fibres in positioning the spindle [16–18]. Retraction fibres contain a variety of specialized membrane, cytoskeletal and matrix-adhesion proteins [11,19–21]. However, the nature of signalling events at retraction fibres, which are important for spindle positioning, remain unclear, hampering our molecular understanding of how retraction fibres determine the plane of cell division.

In polarized epithelia, MARK2/Par1b localizes along the basolateral membrane and mediates the development of membrane domains [22]. Its membrane localization is thought to be negatively regulated by atypical protein kinase C (aPKC)-mediated phosphorylation at T595 [22,23] or T508 [24]. In addition, MARK2 bears a conserved KA1 domain, which in MARK1 and MARK3 is capable of directing the kinase to specific membrane patches [25]. Whether MARK2 has multiple modes of interaction at the plasma membrane and whether its membrane localization is responsive to cytoskeletal changes need to be determined to explain how the kinase monitors and regulates spindle positions.

Here, we show how MARK2's membrane localization is dynamically regulated in mitotic and interphase cells using several live-cell imaging techniques: fluorescence recovery after photobleaching (FRAP) and total internal reflection fluorescence (TIRF) imaging in millisecond time scales along with long-term time-lapse microscopy. We report a novel localization for MARK2 kinase at actin-rich mitotic retraction fibres, which is dependent on its kinase activity. This is different from MARK2's localization at centrosomes, which does not require MARK2's kinase activity. Using a kinase-dead (KD) mutant of MARK2, we uncover a novel role for MARK2 in monitoring cortical actin stress fibres in interphase. We show that MARK2 is required to recentre spindles that are off-centred following actin disassembly, showing the close functional relationship between MARK2 and the actin network. We propose that, during both interphase and mitosis, MARK2 localizes at specialized membrane subdomains and coordinates actin and microtubule cytoskeletal changes, thus enabling normal cell division.

## 2. Results

### 2.1. MARK2 found at membrane subdomains is sensitive to actin status

During interphase, MARK2 localizes along the plasma membrane in non-polarized epithelial cells and specifically basolateral membrane in polarized epithelial cells [22,24,26], but its underlying regulation is not fully understood. To study whether MARK2's activity can influence its membrane localization, we generated HeLa FRT/TO cell lines that conditionally express an RNAi resistant form of MARK2 wild-type (WT) or one of two MARK2 point mutants: MARK2 D157A (KD mutant) or MARK2 T595E (a point mutant to mimic aPKC phosphorylation proposed to block its membrane localization; [26]) (figure 1a). Controlled expression of all three forms of MARK2 (WT, KD and T595E mutant) fused to YFP could be achieved following doxycycline treatment (figure 1b). Live-cell imaging of interphase HeLa FRT/TO cells expressing MARK2-YFP showed that both the KD mutant and the T595E point mutant enrich at the plasma membrane similar to MARK2-WT protein (figure 1c). Comparing deconvolved single-plane

images of three-dimensional (3D) stacks showed localization of YFP-tagged mutants as discontinuous punctate patches at the cell–substrate interface ($Z_{CS}$) and a relatively continuous cortical membrane localization in deeper Z-slices ($Z_3$; 1.5 µm above the substrate), confirming membrane localization in all three forms of MARK2. Thus, the phosphorylation of MARK2 at T595 by aPKC may be insufficient to displace MARK2 from the plasma membrane.

To study the regulation of MARK2-YFP localization at the plasma membrane, we performed TIRF imaging using a super-resolution microscope OMX-SR$^{TM}$. MARK2-YFP (WT) is present at the plasma membrane as dynamic submicrometre-sized domains (figure 1d; see electronic supplementary material, figure S1a and movie S1 for time-lapse images). The MARK2-KD mutant localized as prominent long striations aligned juxtaposed to actin stress fibres stained using SiR-Actin dye ($n = 13$ cells; figure 1e), suggesting MARK2's ability to monitor cortical actin status. Consistent with this notion, MARK2-YFP signals in WT and T595E mutant cells were observed predominantly as densely arranged foci juxtaposing actin stress fibres; fewer striations were observed ($n = 10$ cells WT; 16 cells T595E) (figure 1e,f). In all three cases, MARK2-YFP peak intensity was approximately 200 nm away from the nearest actin stress fibre signal (figure 1e). We conclude that the MARK2-KD mutant localization pattern during interphase is altered.

Structural studies indicate that MARK2 can dimerize [27] and so we compared MARK2-YFP-WT and -KD localizations in cells with reduced levels of endogenous MARK2 following MARK2 siRNA treatment (electronic supplementary material, figure S1B). As in cells that retained endogenous MARK2 (figure 1e), MARK2-siRNA-treated cells expressing MARK2-YFP-WT displayed predominantly a punctate focus-like membrane distribution, while MARK2-siRNA-treated MARK2-YFP-KD-expressing cells displayed prominent long striations (electronic supplementary material, figure S1C; $n_{cells} = 20$ (WT); $n_{cells} = 22$ (KD)). Thus, despite the reduction in endogenous MARK2 levels, the subcellular localization pattern of MARK2-WT protein and MARK2-KD mutant are strikingly distinct.

In summary, the live-cell studies demonstrate a highly dynamic and regulated membrane localization for MARK2 at the cell–substrate interface in interphase: first, MARK2 enriches at the plasma membrane as dynamically moving submicrometre-sized patches, and, second, its localization pattern adjacent to actin fibres (but not membrane association *per se*) is dependent on its kinase activity.

### 2.2. MARK2 is a retraction fibre component regulated by its activity

During mitosis, MARK2 is needed for correct spindle positioning [4] and the distribution of retraction fibres can influence spindle positioning [17]. Hence, we tested whether MARK2 is enriched at retraction fibres found at the cell–substrate interface. Analysing MARK2-YFP-WT localization in mitotic cells showed YFP signal along retraction fibres that extend out of the mitotic cell at the cell–substrate interface, revealing a novel subcellular localization for MARK2 at retraction fibres. However, the length of YFP signal-bearing fibres was noticeably reduced in cells expressing the KD mutant compared with cells expressing either the MARK2-YFP-WT or -T595E mutant, suggesting reduced

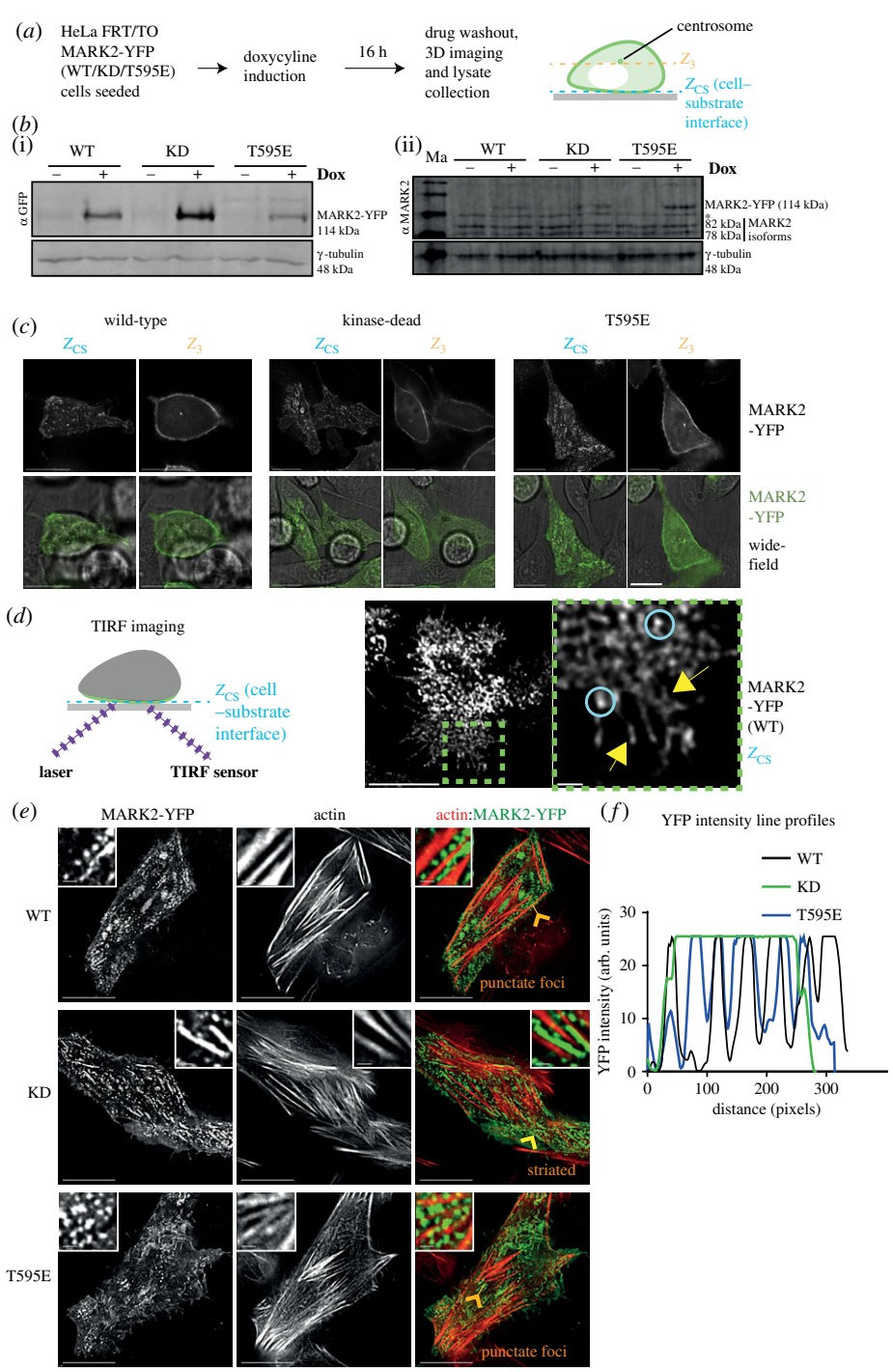

**Figure 1.** MARK2 membrane localization is independent of its kinase activity and Thr595 phosphorylation status. (*a*) Experimental regime: expression of MARK2-YFP (WT) or MARK2-YFP (KD) or MARK2-YFP (T595E) mutants was induced in HeLa FRT/TO cell lines using doxycycline (Dox) for 16 h. Following drug washout, cells were imaged and then lysates collected for immunoblotting. (*b*) Immunoblots showing the conditional expression of MARK2-YFP (WT), MARK2-YFP (KD) or MARK2-YFP (T595E) mutant in HeLa FRT/TO cell lines upon doxycycline treatment. +and − refer to doxycycline (Dox)-treated and -untreated conditions, respectively. Two independent blots probed with antibodies against γ-tubulin and either YFP (i) or MARK2 (ii) show comparable levels of exogenous (MARK2-YFP) and endogenous MARK2 protein expression. *Non-specific band recognized by MARK2 antibody. γ-Tubulin is used as a loading control. 'Ma' refers to marker lane. (*c*) Representative Z-sections from 3D image stacks showing cell−substrate and cell cortex localization of MARK2-YFP (WT), MARK2-YFP (KD) or MARK2-YFP (T595E) mutant as indicated. 'Widefield' refers to white light images acquired to indicate cell periphery. Scale bar: 15 μm. (*d*) Cartoon illustrates TIRF microscopy of MARK2-YFP at the cell−substrate interface. Images on the right (uncropped and cropped as indicated) show MARK2-YFP localization as submicrometre-sized foci (blue rings) or membrane patches (yellow arrows) that move through time (for the rest of the time-lapse images, see electronic supplementary material, figure S1B and movie S1). Scale: 10 μm and insets 1 μm. Cells were treated with doxycycline and control SiRNA 48 h before imaging. (*e*) Deconvolved Z-slice of 3D image stacks show MARK2-YFP signal at the cell−substrate interface in WT, KD or T595E mutant-expressing cells stained with SiR-Actin dye. Yellow arrow indicates prominent 'striated' signal of MARK2-YFP observed adjacent to actin stress fibres in KD mutant-expressing cells. Orange arrows indicate 'punctate foci' of MARK2-YFP adjacent to actin stress fibres in WT and T595E mutant-expressing cells. The predominant localization pattern is described in orange as 'punctate foci' or 'striated' within merged images. Scale as indicated. (*f*) Graph of YFP intensities of lines drawn parallel to actin fibres (as in *e*), using ImageJ software, shows MARK2-YFP fluorescence intensity along distance of line. Line profiles show peaks and troughs along the line in WT or T595E-expressing cells, consistent with 'punctate foci' localization pattern, whereas the KD-expressing cell shows a wider peak length consistent with its 'striated' subcellular localization.

localization of the KD mutant at retraction fibres (electronic supplementary material, figure S2A). We observed a similar difference in the cell–substrate localization pattern in MARK2-WT- and MARK2-KD-expressing cells depleted of endogenous MARK2 (electronic supplementary material, figure S2B), demonstrating that the subcellular localization of MARK2 at retraction fibres is dependent on its kinase activity.

In addition to reduced length of fibres, KD mutant-expressing cells displayed spindles that were frequently tumbled/misoriented as inferred using metaphase plate position (electronic supplementary material, figure S2A). To exclude the possibility of reduced MARK2-YFP-KD localization at retraction fibres arising as an indirect result of spindle tumbling, we treated cells with STLC, an Eg5 inhibitor, and enriched uniformly for monopolar spindles in both WT and mutant-expressing cells (figure 2a) and analysed MARK2-YFP localization at both the mid-cortex region (where spindle poles/centrosomes are visible) and the cell–substrate interface (where retraction fibres are visible) (figure 2b,c). Both the WT and the two point mutants of MARK2-YFP (KD and T595E) were normally enriched at centrosomes and cortical membrane corresponding to the mid-cortex region, as expected (figure 2b). However, MARK2-YFP's membrane localization at the cell–substrate interface was strikingly dissimilar in cells expressing the WT or KD mutant. Deconvolved Z-slices corresponding to the cell–substrate interface ($Z_{cs}$) showed patches of MARK2-YFP-WT signals along retraction fibres that extend out of the rounded up mitotic cell (figure 2c), confirming the novel mitotic localization for MARK2 at the cell–substrate interface, independent of spindle bipolarity status. Quantifying MARK2-YFP signal along the retraction fibres confirmed the reduction in the length of YFP signals at the cell–substrate interface in cells expressing the KD mutant, compared with cells expressing the T595E mutant or WT kinase (figure 2c,d), indicating a role for MARK2's kinase activity in regulating its levels at the retraction fibres. Furthermore, we quantified the proportion of cells with MARK2-YFP signal-bearing retraction fibres at the cell–substrate interface as either 'long' (localization to fibres which extend beyond the mid-cortex boundary of the cell) or 'short' (localization to fibres which are contained within the mid-cortex boundary of the cell). Comparing the proportion of cells with long or short YFP signal at retraction fibres showed that the KD mutant-expressing cells predominantly displayed short MARK2-YFP-bearing fibres at the cell–substrate interface (figure 2e). Taken together, the data show that the membrane localization of MARK2 along retraction fibres, but not the rest of the cortical membrane, is dependent on its kinase activity.

## 2.3. Centrosomal localization of MARK2 is independent of its activity

We investigated whether the centrosome localization of MARK2 is dependent on its kinase activity. To address this, HeLa cells co-expressing MARK2-YFP (KD) mutant and RFP-PACT (RFP fused to pericentrin PACT domain [4]) were treated with MG132 for 60 min to enrich for metaphase cells. MARK2-YFP (KD) colocalized with the RFP-PACT signal, indicating that the mutant can be recruited to the centrosome (figure 3a). To exclude the possibility of indirect

enrichment of MARK2-KD at centrosomes due to vesicular traffic towards spindle poles, cells were incubated with 1.7 µM nocodazole for 30 min to depolymerize all microtubules. The loss of microtubules and the bipolar spindle structure were inferred from the random location of centrosomes in MG132-treated metaphase cells. MARK2-YFP-KD localizes normally at the centrosomes of nocodazole-treated cells (figure 3b). To confirm that the KD MARK2 mutant can localize independent of endogenous MARK2, we studied centrosome localization in MARK2 siRNA-treated HeLa FRT/TO cells expressing an MARK2-siRNA-resistant form of either MARK2-YFP-WT or KD (electronic supplementary material, figure S1B). Z-projections of live-cells showed centrosomal localization of MARK2-YFP WT and KD mutant (figure 3c), confirming that the activity of MARK2 is dispensable for MARK2's localization at the centrosome.

Therefore, neither the activity of MARK2 nor the presence of microtubules is required to localize MARK2/Par1b kinase at the centrosome, highlighting MARK2's intrinsic ability to bind to centrosomes. Thus, MARK2 enrichment at retraction fibres alone, but not the rest of the cortical membrane or centrosomes, is regulated by its kinase activity.

## 2.4. Cortical MARK2 is recruited independent of cortical dynein or microtubules

The loss of MARK2 increases astral microtubule length [4]. We investigated whether the cortical membrane localization of MARK2 is sensitive to cortical dynein or astral microtubule status, which could set a negative feedback loop to control MARK2 levels and maintain microtubule length. This is important to test as MARK2/PAR1 levels at the cortex are dynamically regulated: in *Caenorhabditis elegans*, microtubule binding of PAR2 provides a protected environment to load PAR2 at the cortex, which in turn enriches PAR1 kinase at the cortex [28]. Whether cortical dynein or astral microtubules influence MARK2 localization is not known.

To investigate the extent to which cortical dynein can regulate MARK2 levels at the cell cortex, we depleted the cortical dynein adaptor, LGN, using LGN siRNA oligonucleotides (electronic supplementary material, figure S3A). In LGN siRNA-treated cells, we observed a noticeable reduction in LGN protein levels (electronic supplementary material, figure S3B), but MARK2-YFP was normally localized at the cortical membrane as in control siRNA-treated cells (electronic supplementary material, figure S3C), suggesting that cortical dynein status is not important for MARK2 enrichment at the cortex.

To assess whether MARK2's cortical enrichment is sensitive to microtubules, we used FRAP to bleach a small area of MARK2-YFP at the cortical membrane and measure MARK2-YFP recovery, in the presence and absence of microtubules by treating cells either with DMSO (solvent control) or nocodazole, respectively (figure 4a). FRAP studies showed that MARK2's membrane localization is highly dynamic with a fluorescence recovery rate of 7.07 s (figure 4b,c). There was no significant change in the rate of MARK2-YFP recovery between nocodazole-treated and -untreated cells (figure 4c). We confirmed that the Nocodazole treatment had depolymerized astral microtubules fully by immunostaining using anti-tubulin antibodies (figure 4d). These data together show that astral microtubules do not regulate MARK2 dynamics at the cortical membrane.

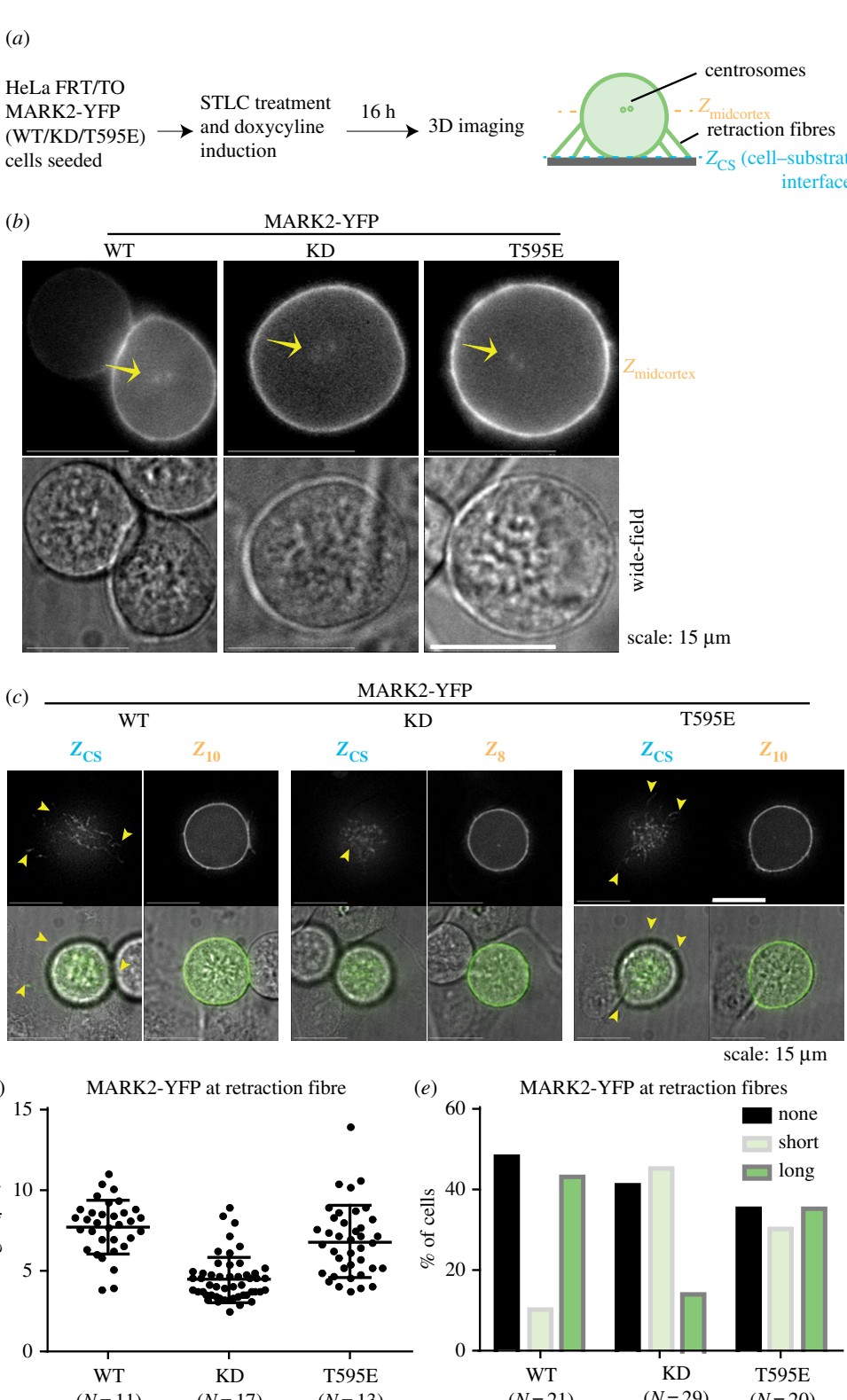

**Figure 2.** MARK2 localization at retraction fibres, but not the rest of the cortex or centrosomes, depends on its activity. (*a*) Experimental regime: expression of MARK2-YFP (WT) or MARK2-YFP (KD) or MARK2-YFP (T595E) mutants was induced in HeLa FRT/TO cell lines using doxycycline. To observe MARK2 localization in mitotic cells, MARK2-YFP-expressing HeLa FRT/TO cells were enriched in prometaphase of mitosis using STLC, an Eg5 inhibitor, which blocks monopolar to bipolar spindle conversion. Cells were imaged soon after STLC washout. (*b*) Representative live-cell still images of HeLa FRT/TO cells expressing MARK2-YFP (WT), MARK2-YFP (KD) or MARK2-YFP (T595E) mutant as indicated. Widefield refers to white light images acquired to indicate the periphery of rounded-up mitotic cells. Yellow arrows refer to separating spindle poles soon after STLC washout. Scale as indicated. (*c*) Representative Z-slices from 3D image stacks showing cell–substrate and cell cortex localization of MARK2-YFP (WT), MARK2-YFP (KD) or MARK2-YFP (T595E) mutant as indicated. Widefield refers to white light images acquired to indicate the periphery of rounded-up mitotic cells. Yellow arrows refer to retraction fibres at the cell–substrate or cell–cell interface. Scale as indicated. (*d*) Graph showing the distribution of YFP signal lengths at retraction fibres in HeLa FRT/TO cells expressing one of the three forms of MARK2 as indicated. *N* refers to the number of mitotic cells. (*e*) Graph of percentage of cells showing the presence (long or short) or absence of MARK2-YFP at retraction fibres assessed using YFP signals in HeLa FRT/TO cells expressing one of the three forms of MARK2 as indicated. 'Long' refers to MARK2-YFP localization to fibres which extend beyond the mid-cortex boundary of the cell and 'short' refers to MARK2-YFP localization to fibres which are contained within the mid-cortex boundary of the cell. *N* refers to the number of mitotic cells.

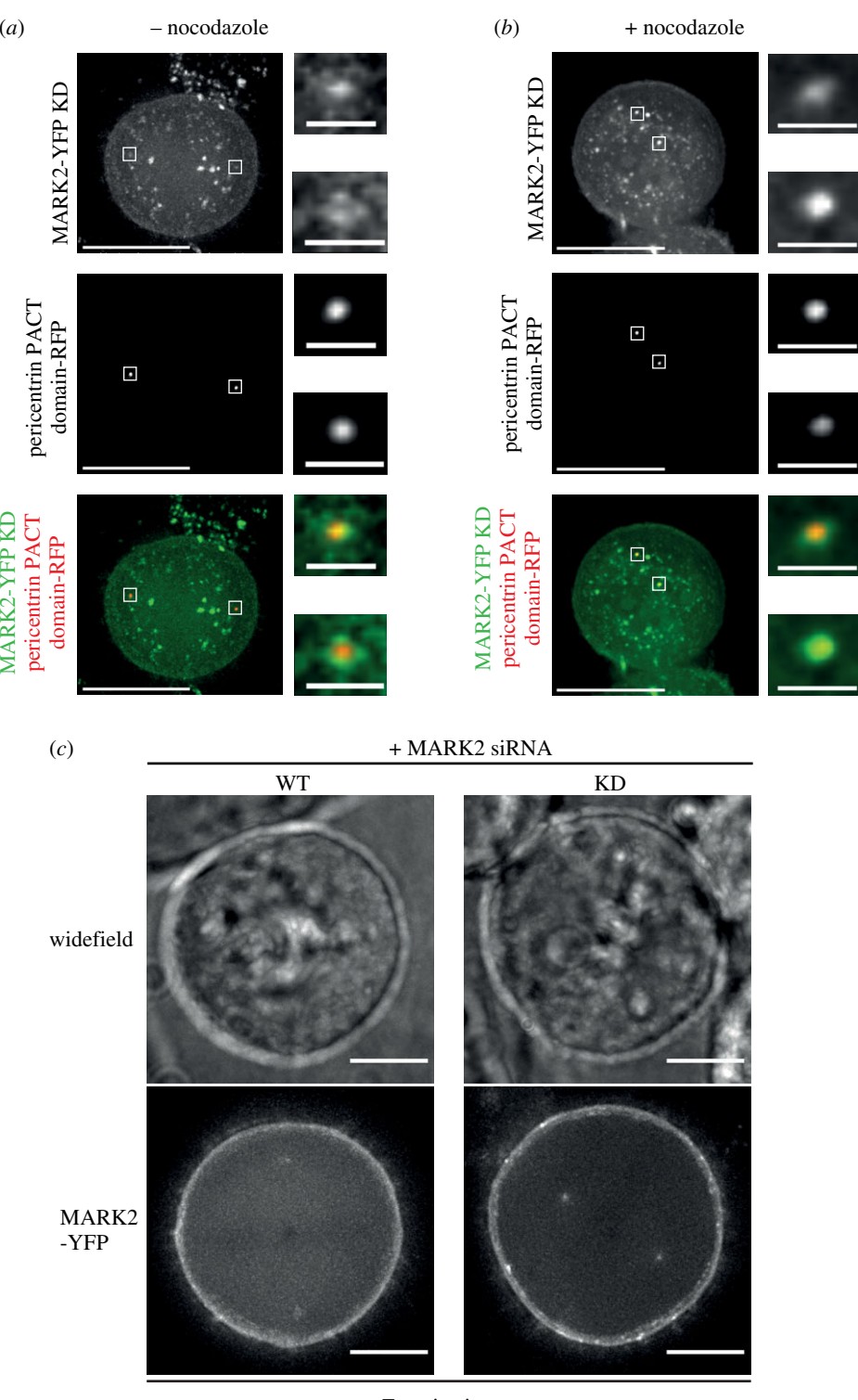

**Figure 3.** MARK2 localization at centrosomes is independent of its activity. (*a,b*) Representative live-cell still images of HeLa cells expressing exogenous MARK2-YFP-KD and the centrosome-localizing PACT domain of pericentrin tagged with RFP to mark centrosomes, illustrating the localization of MARK2-YFP-KD in (*a*) metaphase cells treated with MG132 alone and (*b*) metaphase cells exposed to nocodazole treatment in addition to MG132. Insets show magnified areas of centrosome as indicated by white boxes on main images. Images of cells in metaphase collected after a 60 min treatment with MG132. For nocodazole treatment, cells were exposed to 1.7 μM nocodazole for 30 min after MG132 treatment. Scale: 10 μm in main image and 1 μm in insets. (*c*) Z-projections of representative live HeLa FRT/TO MARK2-YFP-WT or KD cells transfected with MARK2 siRNA and induced to express MARK2-YFP using doxycyline. Maximum intensity Z-projections show normal spindle pole association of both MARK2-YFP-WT and MARK2-YFP-KD mutant following the depletion of endogenous MARK2. Ten slices for WT and seven for KD were used to incorporate both centrosomes of the bipolar spindle. Z-slices are 0.5 μm apart. Scale: 15 μm.

The FRAP studies and LGN depletion studies demonstrate that the cortical enrichment of MARK2 is insensitive to the status of astral microtubules and cortical dynein—two established key regulators of spindle movements. These findings position MARK2 as an independent upstream regulator of the spindle-positioning process.

## 2.5. MARK2 but not dynein is enriched at actin-rich retraction fibres

To compare the localization of cortical dynein and MARK2 kinase, we performed deconvolution live-cell microscopy of SiR-Actin-stained HeLa FRT/TO cells expressing

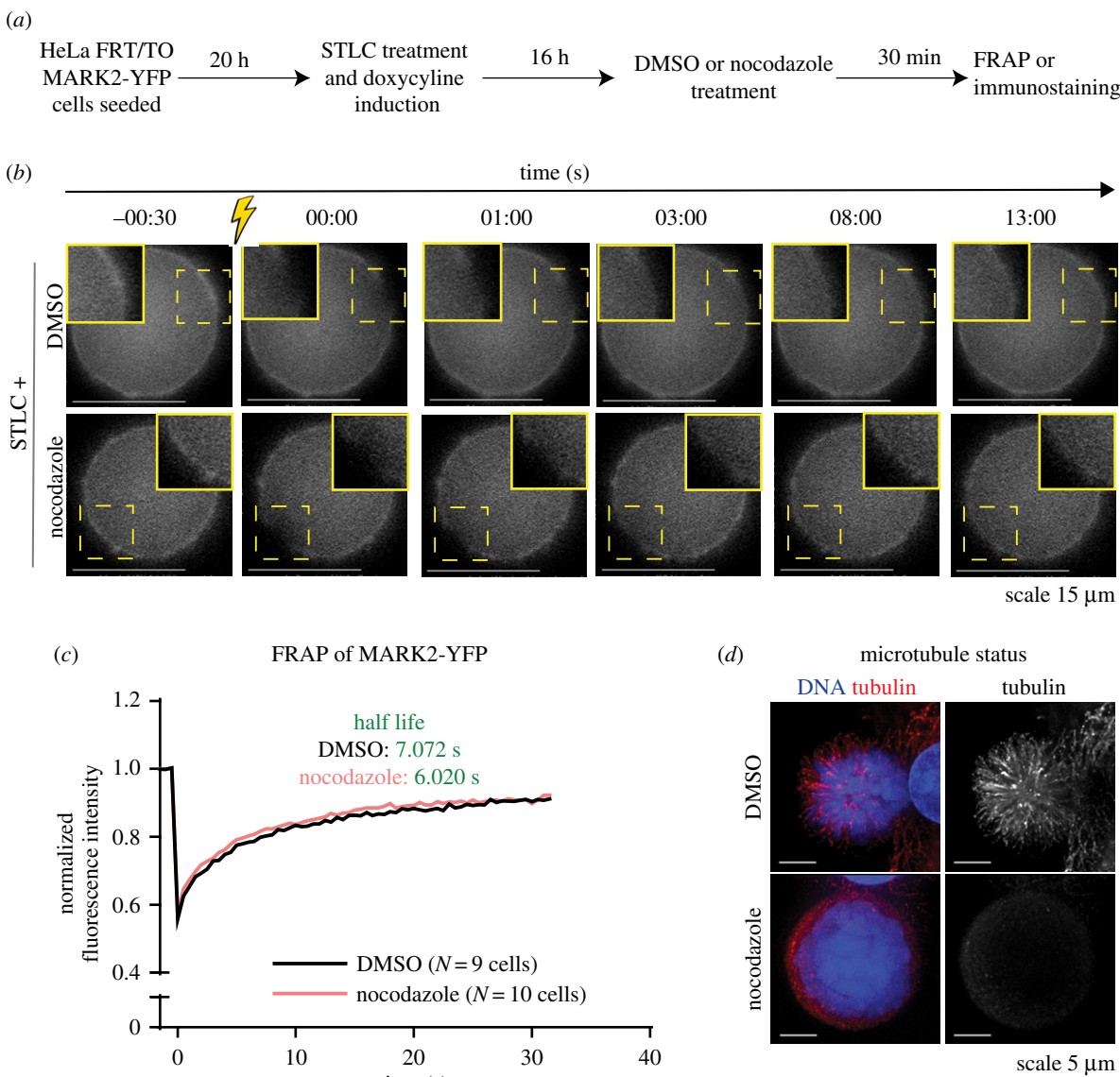

**Figure 4.** MARK2 localization at the mitotic cortex is dynamic and independent of microtubules. (*a*) Experimental regime—HeLa FRT/TO MARK2-YFP-WT cells were seeded. MARK2-YFP expression was induced with doxycycline and cells arrested at prometaphase using STLC. Sixteen hours later, cells were treated with STLC and either DMSO or nocodazole (as indicated) was added prior to FRAP or immunostaining. (*b*) Images of cells (treated as in *a*) before and after FRAP. Yellow square indicates bleached area. (*c*) Quantification of FRAP intensities in cells treated as in (*a*) and shown in (*b*). Background values were subtracted and values were normalized to opposing cortical values, as well as pre-bleach values (collated from two sets). Each cell was bleached at three cortical areas and recovery of fluorescence taken at each site. FRAP rates were obtained using a nonlinear fit plateau followed by one-phase decay. (*d*) Representative deconvolved images show nocodazole-mediated disruption of microtubules in HeLa FRT/TO MARK2-YFP cells treated as in (*a*) and immunostained with antibodies against tubulin and stained with DAPI for DNA. Images are maximum intensity Z-projections. Scale: 5 μm.

MARK2-YFP (WT) and HeLa cells expressing dynein heavy chain fused to GFP (DHC-GFP). As expected, mitotic cells expressing MARK2-YFP showed a clear localization of MARK2-YFP and SiR-Actin signals along the retraction fibres in Z-slices close to the cell–substrate interface (blue arrowheads; figure 5*a*). At upper Z-slices, away from the cell–substrate interface, cortical MARK2-YFP signal was frequently observed along the outer surface of cortical SiR-Actin signal (figure 5*a*), consistent with MARK2's membrane localization. Unlike MARK2-YFP, no DHC-GFP signal was observed at retraction fibres in HeLa cells expressing DHC-GFP. At higher Z-sections, where the spindle signal of DHC-GFP was visible, cortical DHC-GFP signal was frequently observed along the inner surface of cortical actin signal (purple arrowhead; figure 5*b*). Based on the 3D image stacks, we conclude that MARK2 but not dynein is found at the retraction fibres, and that cortical dynein and

MARK2 occupy the inner and outer layers of the cell cortex, respectively (figure 5*c*).

## 2.6. Retraction fibres form independent of MARK2

We find that MARK2 is recruited to retraction fibres in a kinase activity-dependent manner; whether MARK2 is needed for the formation or maintenance of retraction fibres is not known. To address this, we assessed retraction fibres using a membrane marker, CellBrite™. To deplete MARK2, previously standardized siRNA oligonucleotides against MARK2 were used (figure 6*a*) [4]. Both control siRNA-treated cells and MARK2 siRNA-treated mitotic cells displayed membrane signal corresponding to retraction fibres at the cell–substrate and cell–cell interface (figure 6*b*). We conclude that MARK2 is not essential for the formation or maintenance of mitotic retraction fibres.

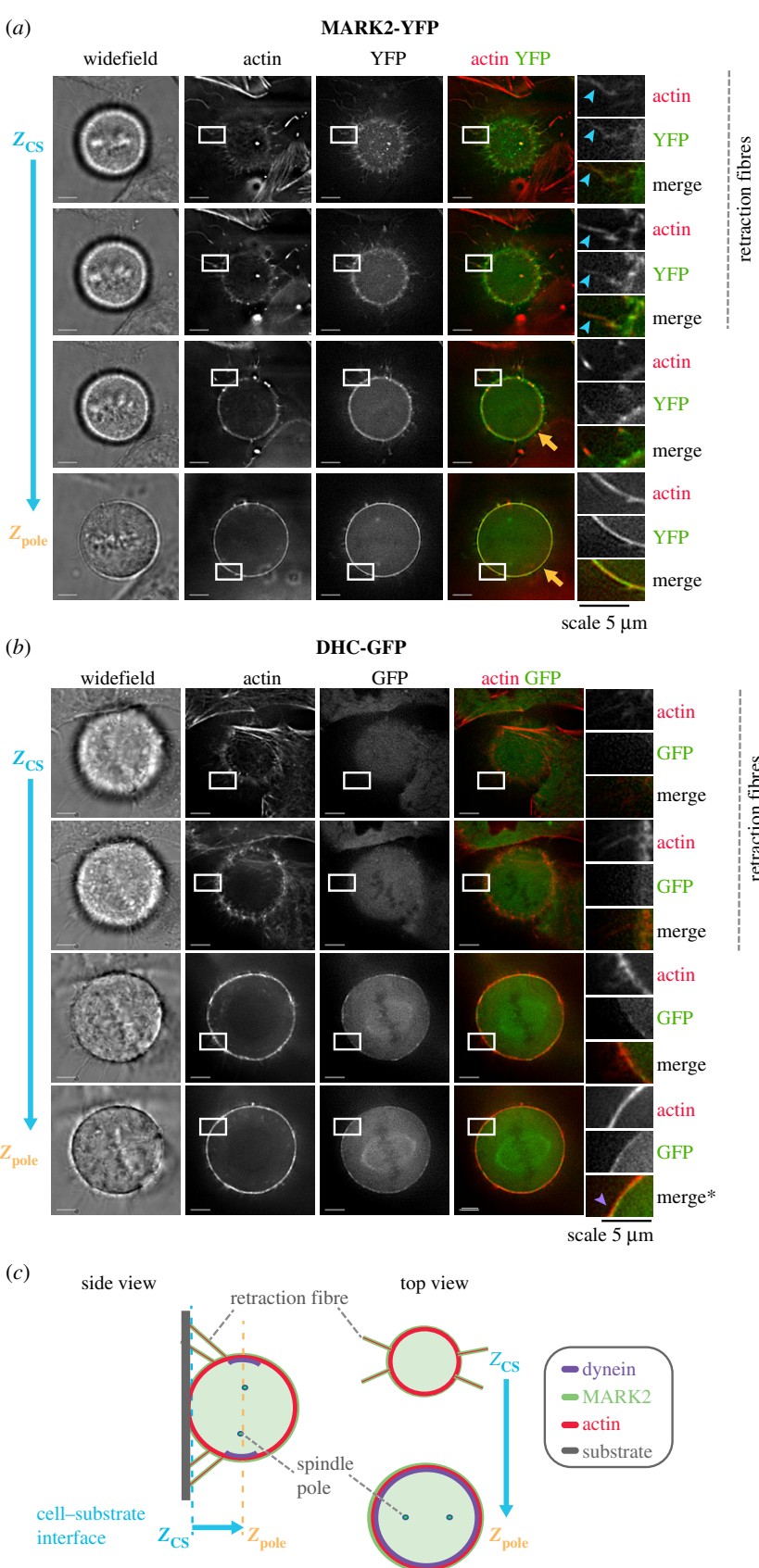

scale 5 μm

**Figure 5.** MARK2 and dynein occupy distinct areas of the mitotic cell cortex. (*a*) Live-cell deconvolved single-plane images of a doxycycline-treated HeLa FRT/TO cells expressing MARK2-YFP (WT) stained with SiR-actin dye for 30 min prior to imaging. Two Z-slices of 3D image stacks close to the cell–substrate interface (Z$_{cs}$) show the presence of MARK2-YFP signal at retraction fibres marked using SiR-Actin dye (in red). Z-slices of 3D image stacks near the spindle pole position (Zpole) show MARK2-YFP signal (in green) enveloping cortical actin signal (in red; marked using yellow arrows). Blue arrowheads mark retraction fibres in cropped images. Scale as indicated. (*b*) Live-cell deconvolved single-plane images of a HeLa cell expressing mouse DHC-GFP from an endogenous promoter stained with SiR-actin dye for 30 min prior to imaging. Two Z-slices of 3D image stacks close to the cell–substrate interface (Z$_{cs}$) show the absence of DHC-GFP signal at retraction fibres marked using SiR-Actin dye (in red). Z-slices of 3D image stacks near the spindle pole position (Z$_{pole}$) show cortical actin signal enveloping DHC-GFP signal (in green; purple arrowhead; merge crop marked *). Scale as indicated. (*c*) Cartoon showing the relative positioning of Z-slices (i) at cell–substrate interface (Z$_{cs}$) where retraction fibres are visible and (ii) above (Z$_{pole}$) where the two spindle poles are visible. Both side view and top view of 3D image stacks presented. Top view images illustrate the presence of cortical dynein signal at Z$_{pole}$ but not Z$_{cs}$.

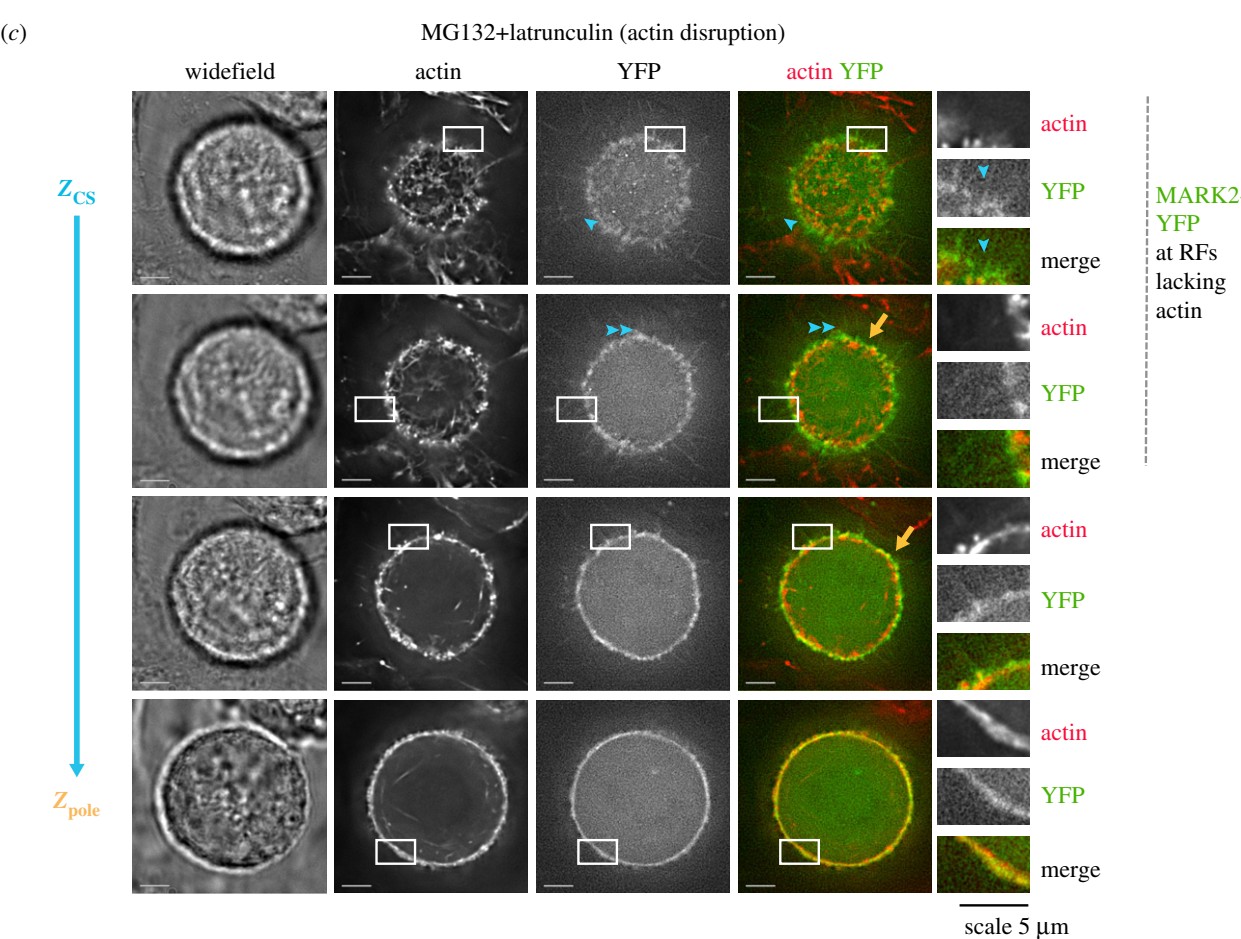

**Figure 6.** Retraction fibres form independent of MARK2, and MARK2 localizes at fibres independent of actin. (*a*) Immunoblot shows the extent of depletion of MARK2 following the treatment of cells with control or MARK2 siRNA as indicated using antibodies against MARK2 (upper image) or γ-tubulin (lower image). γ-Tubulin is used as a loading control. (*b*) Representative images of cells treated with control or MARK2 siRNA, as indicated, were fixed and stained with CellBrite™ (membrane marker) and DAPI (DNA marker). Single-plane images shown are Z-slices corresponding to the cell–substrate interface extracted from deconvolved 3D image stacks. Orange arrows mark SiR-membrane-stained retraction fibres that stretch out of the rounded-up cell cortex. Scale: 15 μm. (*c*) MARK2 localization at retraction fibres is independent of actin: live-cell images of doxycyline-treated HeLa FRT/TO cells expressing MARK2-YFP (WT) exposed to latrunculin A and SiR-actin dye for 30 min prior to imaging. Z-slices of 3D image stacks corresponding to spindle pole position show MARK2-YFP signal enveloping cortical actin signal (orange arrows) and Z-slices of 3D image stacks corresponding to the cell–substrate interface show MARK2-YFP signal (in green) at retraction fibres lacking SiR-actin signal (in red). Blue arrowheads mark retraction fibres with MARK2-YFP (corresponding actin image shows loss of actin following latrunculin A treatment). Compare figure 4*a* for latrunculin-untreated cells. Scale: 5 μm.

As retraction fibres form normally in the absence of MARK2, we investigated whether the disruption of actin affects MARK2 localization at retraction fibres. To disrupt actin in metaphase, we treated cells with low doses of latrunculin and MG132 and studied the localization of MARK2 in retraction fibres that lacked SiR-Actin signals. In retraction

royalsocietypublishing.org/journal/rsob    Open Biol. 9: 180263

fibres that lacked SiR-Actin, we did not observe any stark reduction in MARK2-YFP signal (figure 6c), indicating that MARK2-YFP recruitment at retraction fibres is independent of actin status at retraction fibres.

## 2.7. MARK2 corrects spindle off-centring caused by actin disassembly

MARK2 regulates microtubule length and it ensures normal equatorial centring of spindles [4]. In addition to microtubules, an intact actin network and proper cross-linking of actin filaments to microtubules are also required for normal positioning of spindles [21,29–31]. Because MARK2 localizes independently of actin at retraction fibres, we asked whether actin disassembly-induced spindle off-centring is monitored and corrected by MARK2. To address this, we studied equatorial spindle off-centring and recentring in cells treated with low doses of latrunculin in the presence and absence of MARK2 (electronic supplementary material, figure S4A,B). Spindle off-centring and recentring events were monitored in HeLa cells co-expressing mCherry-tubulin and histone 2B-GFP using time-lapse images acquired once every 4 min [9]. As expected, time-lapse movies of cells treated with control siRNA, but not MARK2 siRNA, showed rapid recentring of equatorially off-centred spindles within 4−8 min (figure 7a). We next quantified the percentage of spindles that underwent off-centred to centred (OC to C) correction episodes within an 8 min time window (figure 7b). Nearly 73% of control siRNA-treated cells showed at least 50% success in correcting off-centred spindles (OC to C episodes within 8 min) (figure 7b). However, only 29% of MARK2 siRNA-treated cells showed at least 50% success in correcting off-centred spindles (OC to C episodes within 8 min) (figure 7b), although spindle recentring at anaphase was nearly 80% (electronic supplementary material, figure S4C), confirming MARK2's pre-anaphase role in equatorial centring of spindles [4]. This is consistent with differential control of anaphase and pre-anaphase spindle positioning events (reviewed in [32]) and also different sets of microtubule plus-end binding proteins in the two phases [33].

To assess whether MARK2 plays a role in correcting spindle off-centring events induced by actin disassembly, cells were exposed to a low dose of latrunculin 30 min prior to imaging (electronic supplementary material, figure S4A). In control siRNA-treated cells, exposure to latrunculin induced an increased incidence of spindle off-centring (figure 7a; electronic supplementary material, movie S2), showing the need for an intact actin network for stable equatorial centring of spindles. Nevertheless, 80% of control siRNA-treated cells exposed to latrunculin showed at least 50% success in correcting the equatorially off-centred spindles (OC to C episodes within an 8 min time window), suggesting an underlying mechanism to monitor and correct spindle off-centring induced by actin disassembly (figure 7c). We next analysed MARK2 siRNA-treated cells exposed to latrunculin: the number of cells that underwent mitosis was reduced ($n = 21$ cells, compared with 60 cells in control siRNA-treated cells exposed to latrunculin) and a mild congression defect was observed in approximately 15% of cells. Strikingly, less than 5% of MARK2 siRNA-treated cells exposed to latrunculin showed at least 50% success in OC to C episodes within 8 min (figure 7c; electronic

supplementary material, movie S3). These data highlight MARK2's role in correcting spindle off-centring induced by actin perturbation. A large proportion of off-centred spindles in MARK2-depleted cells underwent centring at anaphase transition (electronic supplementary material, figure S4D), highlighting MARK2's pre-anaphase role in correcting spindle off-centring induced by actin perturbation.

To quantify the efficiency of the correction process, we next compared the rates of OC with C episodes in the presence and absence of MARK2 in latrunculin-treated cells. A cumulative frequency graph showed that 70% of OC to C episodes were completed within either 8 or 16 min following control siRNA or MARK2 siRNA treatment, respectively (electronic supplementary material, figure S4E), showing a slight delay in OC to C rates following MARK depletion. In stark contrast, OC to C episodes were very delayed in MARK2 siRNA-treated cells exposed to latrunculin: 70% of OC to C episodes were completed within 50 min, compared with 25 min in control siRNA-treated cells. We conclude that MARK2 kinase plays a key role in correcting spindle off-centring induced by actin perturbation.

## 3. Discussion

Here, we report MARK2/Par1b, a known regulator of microtubule stability, as a novel component of retraction fibres with a role in correcting spindle off-centring induced by actin disassembly. Its recruitment to retraction fibres alone, but not the rest of the mitotic cortical membrane, is dependent on its kinase activity. Importantly, its dynamic localization at the mitotic cell cortex is independent of cortical dynein, astral microtubules and the actin network, highlighting its upstream position among the regulators of spindle movements. During interphase, MARK2/Par1b is enriched at the interphase plasma membrane as submicrometre-sized punctate patches that are highly mobile and coincident with actin stress fibres in a kinase activity-dependent manner. We propose that MARK2 recruitment to specialized membrane subdomains can be regulated to monitor and mediate localized cytoskeletal changes in both mitosis and interphase.

MARK2 regulates microtubule dynamics during both interphase and mitosis [4,34–38]. We had shown that spindle centring defects observed in MARK2-depleted cells can be rescued by stabilizing the microtubules [4], indicating a role for MARK2 in spindle pushing forces. However, it was unclear how a cortex- and spindle pole-localized MARK2 can precisely regulate astral microtubule length to achieve equatorial spindle centring. Here, we report that MARK2 localizes at retraction fibres in a kinase activity-dependent manner. This opens a new paradigm for MARK2 activating/inactivating enzymes (phospho-regulation of MARK2) to locally regulate microtubule dynamics in actin-rich areas (retraction fibres; figure 7a). Such localized regulation is crucial as microtubule and actin network changes have to be spatially and temporally coordinated during mitosis (reviewed in [39]). Localized microtubule-regulatory mechanisms may act in addition to the well-established cortical dynein-mediated spindle-positioning pathways as described in C. elegans [40].

MARK2 can dimerize in vitro [27]. To exclude the possibility of endogenous MARK2 influencing the subcellular localization of MARK2-KD by forming dimers, we compared KD localization in both MARK2 siRNA-treated and -untreated

royalsocietypublishing.org/journal/rsob    Open Biol. 9: 180263

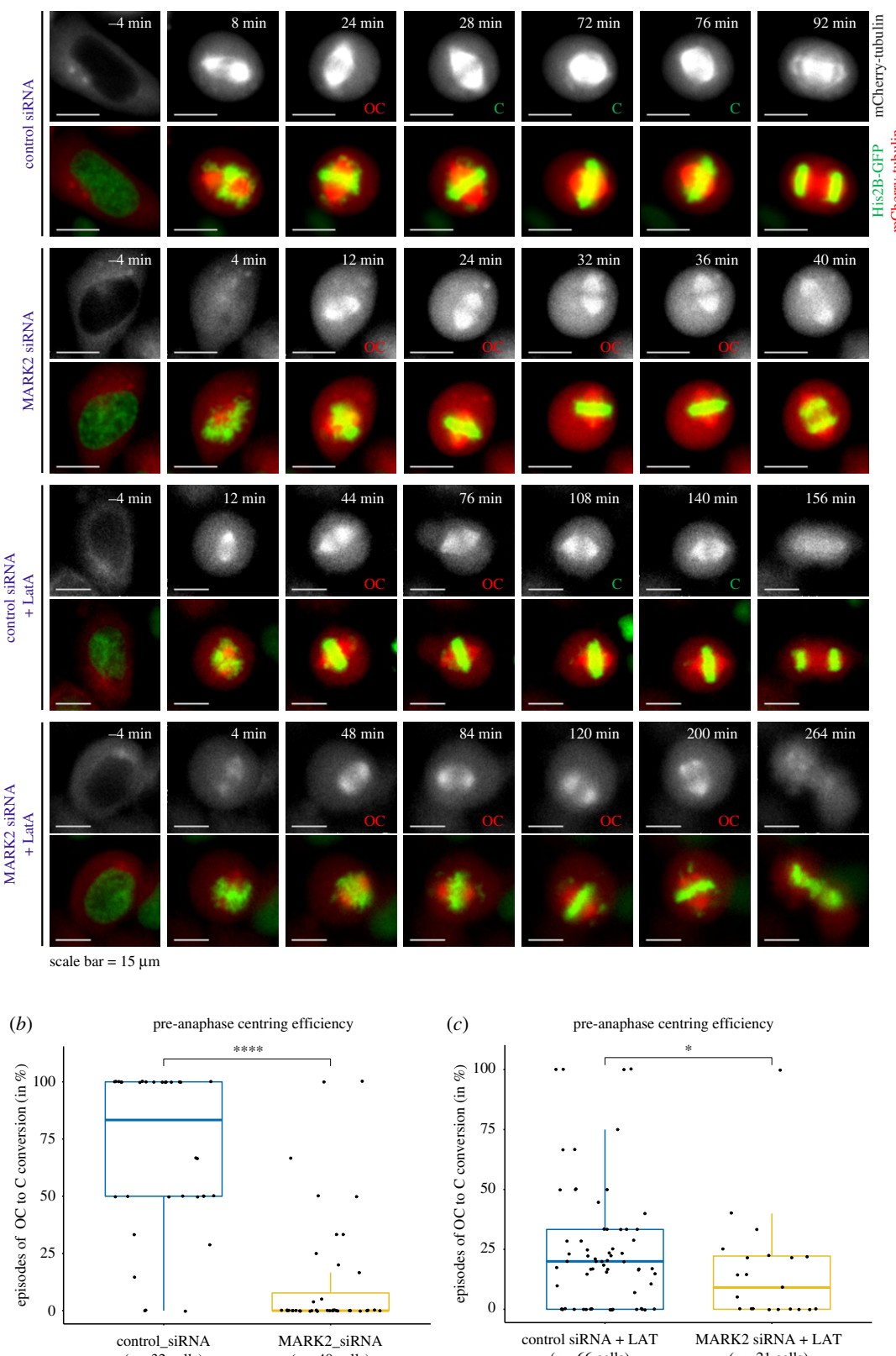

**Figure 7.** MARK2 is needed to correct spindle off-centring induced by actin disassembly. (a) Representative time-lapse image stills of HeLa (His-GFP; mCherry-tubulin) cells transfected with siRNA as indicated. One hour prior to live-cell imaging, the actin perturbant latrunculin A (Lat-A) was added and time-lapse imaging performed for 5 h. OC refers to off-centred spindles (assessed on the basis of equatorial spindle position within the cell boundary (inferred from mCherry-tubulin signals on the spindle and the cytoplasm). Scale as indicated. (b,c) Box plot shows percentage of episodes of equatorially off-centred spindles that centred within an 8 min time window. For pair-wise comparison, a non-parametric Wilcoxon signed-rank test was performed. Asterisks indicate statistical significance for $p \leq 0.05$. N values indicate the number of cells from at least three independent repeats.

cells. In both MARK2 siRNA-treated and -untreated interphase cells, the MARK2-KD mutant localizes as striations, whereas WT localizes predominantly as dispersed foci.

Similarly, in both MARK2 siRNA-treated and -untreated mitotic cells, we find that MARK2-KD localization is altered along retraction fibres and induces spindle misorientation/

royalsocietypublishing.org/journal/rsob     Open Biol. 9: 180263

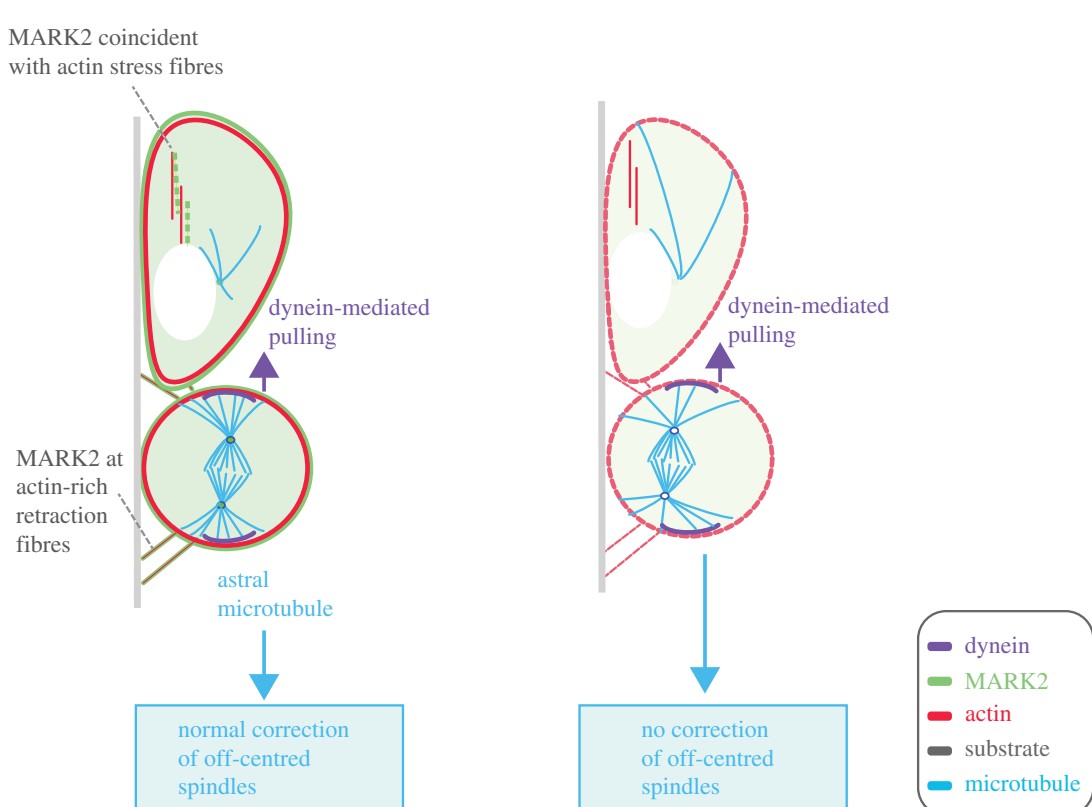

(a) regulated MARK2 localization

MARK2 coincident with actin stress fibres

dynein-mediated pulling

MARK2 at actin-rich retraction fibres

astral microtubule

normal correction of off-centred spindles

(b) actin disassembly and MARK2 loss

dynein-mediated pulling

no correction of off-centred spindles

- dynein
- MARK2
- actin
- substrate
- microtubule

**Figure 8.** Illustration of MARK2-mediated recognition of actin status and regulation of spindle position. Cortical MARK2 and dynein occupy outer and inner cortical areas, in alignment with their distinct roles in spindle centring and pulling, respectively. MARK2 recognizes actin status in interphase (a), and actin perturbation-induced spindle off-centring can be corrected by MARK2 (b). To enable spindle centring, we propose that MARK2 at the cortical membrane can selectively act on astral microtubule ends that reach retraction fibres.

tumbling. These studies show that the MARK2-KD mutant can dominantly interfere with both interphase and mitotic localization and function of MARK2, highlighting the importance of MARK2's kinase activity. However, it is important to add that in cases where MARK2-KD did not show any difference in localization compared with MARK2-WT—for example, centrosomal localization—it is difficult to exclude the contribution of traces of endogenous MARK2 supporting the centrosomal localization of MARK2-KD. Addressing this may require future deletion-mutant studies of the precise centrosome-docking domain in MARK2.

Our findings are consistent with a model where MARK2 dynamically links actin and microtubule cytoskeletal status to regulate spindle movement independent of cortical dynein (see below). Although cortical dynein is important for spindle positioning [41], cortical dynein-independent steering of aster movement has been reported in *C. elegans* [42,43]. We reported a cortical dynein-independent role for MARK2 in equatorial spindle positioning in human cells [4], but the upstream cues remained unclear. Here, we find that MARK2's localization juxtaposing actin stress fibres is regulated by its kinase activity, indicating the kinase can monitor actin status. Moreover, MARK2-mediated phosphorylation of Tau can attenuate Tau binding to F-actin and in turn remodelling the cytoskeletal network [44]. We propose that MARK2 at actin-rich retraction fibres is well positioned to correct off-centred spindles by linking cortical microtubule and actin status (figure 8a). In the absence of MARK2, actin disassembly induces spindle off-centring that

cannot be corrected normally (figure 8b). Although we were unable to perform rescue studies following actin disassembly, a quantitative comparison of pre-anaphase and anaphase spindle movements (figure 7c; electronic supplementary material, figure S4D) shows MARK2's correction role to be particularly important for pre-anaphase stages of mitosis. This pre-anaphase role for MARK2 in spindle recentring is independent of cortical dynein function for at least two reasons: first, MARK2 and dynein heavy chain occupy different regions of the cell cortex (figure 8a). Second, in the absence of cortical dynein (LGN depletion), neither MARK2 localization (in this study) nor spindle centring is lost [4].

We have recently shown that targeting specific kinases and phosphatases in a spatially regulated manner can regulate the interaction of microtubules at chromosome-microtubule attachment sites [45]. Similarly, targeting of kinases to specific membrane domains is thought to contribute to 'coincidence detection', a mechanism that allows kinases to mediate events in a localized fashion [25]. Spatial exclusivity is known to play an important role in mitotic signalling [46]. In the case of MARK2, a coincidence detection mechanism can operate at retraction fibres as MARK2 localization is controlled by its own kinase activity, MARK2 levels at the cortex are independent of actin or microtubule status (reported here) and MARK2-interacting PTPRF phosphatase is a component of retraction fibres forming independent of F-actin [20,47]. Although a single-site phosphorylation at Thr595 [26] is insufficient to fully block MARK2 recruitment to membrane patches, it is possible that alternate or multiple

sites on MARK2 need to be phosphorylated by aPKC for 14–3–3 interaction and its delocalization from membrane. For example, an alternate site T508 when mutated to Ala increases MARK2's membrane localization [24]. In the case of xPar-1, xPKC phosphorylates multiple sites—Thr593 and Ser646 [48], and in the case of closely related MARK3, aPKC phosphorylation at 17 sites is needed to fully abrogate 14–3–3 interaction [24]. Regulating MARK2's membrane localization can be an important signalling gateway for external cues to locally control MARK2-mediated cytoskeletal changes during cell division.

# 4. Material and methods

## 4.1. Cell culture and synchronization

HeLa cells were cultured in DMEM supplemented with 10% fetal calf serum (FCS) and antibiotics (penicillin and streptomycin) [49] and plated onto glass-bottomed dishes (LabTek) or 13 mm round coverslips for imaging. As indicated in the text, to synchronize cells at the metaphase–anaphase transition, cells were treated with 10 μM MG132 (1748; Tocris). Cells were synchronized using either a single 1 μg ml$^{-1}$ aphidicolin block for 24 h and then released for 7 h before filming or by exposing cells to 20 μM S-trityl-L-cysteine (STLC) for 16 h prior to imaging. To disassemble microtubules, cells were treated with 1.7 μM nocodazole (487928; Thermo Fisher Scientific).

## 4.2. Cell line generation

The HeLa FRT/TO cell line expressing siRNA-resistant MARK2-YFP-WT or KD mutant was generated by transfecting a Tet-inducible expression vector encoding siRNA-resistant MARK2-YFP-WT or KD and followed by colony picking [4]. Vectors bearing point mutants of MARK2 were generated by polymerase chain reaction-based point mutagenesis and confirmed by DNA sequencing. Cell line generation procedures were followed according to the FRT/TO system protocol (Invitrogen). Doxycycline induction was usually performed for 16 h prior to imaging (unless differently specified). An amount of 200 ng ml$^{-1}$ doxycycline was used in conventional DMEM supplemented with 10% FCS. The HeLa DHC-GFP cell line expressing mouse DHC-GFP through an exogenous promoter is a kind gift from the Hyman laboratory. The HeLa His2B-GFP mCherry-tubulin cell line was generated as described in [50].

## 4.3. Plasmid and siRNA transfections

HeLa cells were transfected with either siRNAs or plasmid vectors as described previously [51]. siRNA oligonucleotides used to deplete MARK2 were used as standardized previously [4]: MARK2-1 (5′-CCUCCAGAAACUAUUCCGC GAAGUA-3′) and MARK2-2 (5′-UCUUGGAUGCUGAUAU-GAACAUCAA-3′), and LGN [9]. siRNA oligonucleotides were purchased from GE Healthcare. Sequences of plasmid vectors are available upon request.

## 4.4. Live-cell time-lapse imaging and analysis

High-resolution TIRF with deconvolution images were acquired using OMX-SR$^{TM}$. Control siRNA-treated cells were treated with doxycyline for 16–20 h prior to imaging. Images were acquired every second for 20 s, with three z-stacks each 0.125 nm apart, and deconvolved. Time points were equalized.

FRAP was performed on a Deltavision Core$^{TM}$ microscope using Quantifiable Laser module components (488 nm laser). The target site was bleached with a pulse duration of 1 s and laser power of 20%. Three pre-bleach images were acquired 0.5 s apart, and 64 post-bleach images were acquired 0.5 s apart. Time 0 refers to the first post-bleach image acquired. Fluorescence intensities at the bleach spot and an opposing cortical site were measured using ImageJ and analysed as in [52]. Briefly, values were corrected to the ratio between opposing cortical sites at each time point relative to pre-bleach values (to account for acquisition-associated photobleaching), and normalized relative to pre-bleach values. Halftimes were calculated using nonlinear regression analysis (plateau followed by one-phase decay) using Prism Graphpad.

Cells were transfected with siRNA oligonucleotides or plasmid vectors, for 72 or 24 h, respectively (72 h for MARK2, control and LGN siRNA). Before imaging, cells were transferred to Leibovitz L15 medium (Invitrogen) for imaging at 37°C. To observe chromosome and spindle movements, images were acquired with exposures of 0.1 s from at least three Z-planes 3 μm apart every 4 min for 5 h using a 40 × 0.75 NA objective on a DeltaVision Core$^{TM}$ microscope (GE Healthcare) equipped with a Cascade2 camera under EM mode. For colocalizing DHC-GFP, MARK2-YFP and SiR-Actin signals, live cells were imaged (at least 15 Z-slices, 0.5 μm apart) using a 100 × 1.2 NA objective on the microscope described above.

For MARK2 (centrosome) colocalization studies, the centrosome marker (a fragment of pericentrin tagged to RFP) was used (kind gift from J. Pines, ICR, London). Time-lapse videos were analysed manually using Soft-WoRx$^{TM}$. Spindles in HeLa (His-GFP; mCherry-tubulin) cells were visually scored as equatorially off-centred when unequal distances were observed between the cell cortex and the two opposing edges of the metaphase plate (histone-GFP signal) or the cell cortex and the two walls of the spindle at the equator (mCherry-tubulin signal) as in [4].

## 4.5. Immunofluorescence and immunoblotting

For immunofluorescence, the anti-tubulin antibody (1 : 1000, Abcam, ab6160) was used. For membrane visualization, the membrane dye (Biotium, CellBrite$^{TM}$, 30023) was incubated on cells for 15 min prior to fixation, as per the manufacturer's instructions. DNA was stained with DAPI. Images of immunostained cells were acquired using a 100 × 1.2 NA objective on a DeltaVision Core microscope equipped with a CoolSnap HQ Camera (Photometrics).

For immunoblotting, antibodies against γ-tubulin (1 : 800, Sigma-Aldrich, T6793), GFP (1 : 1000, MBL, 598), LGN (1 : 1000, Bethyl Lab, A303-032A) and MARK2 (1 : 1000, Novus, NH00002011-M01) were used. Immunoblots were developed using fluorescent secondary antibodies (LI-COR Biosciences),

royalsocietypublishing.org/journal/rsob   Open Biol. 9: 180263

and fluorescent immunoblots were imaged using an Odyssey imager (LI-COR Biosciences).

## 4.6. STLC wash-off assay

Cells were treated with 20 µM STLC for 16 h to enrich for mitotic cells. Prior to imaging, media were changed and STLC washed off with at least four quick washes in Leibovitz medium.

## 4.7. Latrunculin treatment

For live-cell imaging in figure 6, cells were treated with 1 µM latrunculin A and 10 µM MG132 in Leibovitz medium and imaged 30 min later. For live-cell imaging in figure 7, 100 nM latrunculin A was added prior to imaging.

## 4.8. Statistical analysis

RStudio Software (v. 1.1.456) with R distribution (v. 3.5.1) for statistical computing and GraphPad Prism 5.0 (GraphPad Software, Inc., San Diego, CA, USA) were used to generate graphs and perform statistical analysis. Visual and analytical assessment of normality and homogeneity of variance was performed with the QQ plot and Shapiro–Wilk normality test. The choice for non-parametric tests is in alignment with the underlying data structure. To determine the statistical significance of differences between the population mean ranks in the experimental conditions, a Wilcoxon signed-rank test was used as indicated in the figure legends. Box plots show the median, upper (75%) and lower (25%) quantiles as the box, whiskers represent 1.5 times the interquartile range, and the remaining points are shown as outliers. All data points were included in statistical analyses.

The following convention for asterisk symbols indicating statistical significance was used: (ns) for $p > 0.05$, (*) for $p \leq 0.05$, (**) for $p \leq 0.01$, (***) for $p \leq 0.001$ and (****) for $p \leq 0.0001$. For OC to C rates in MARK2 and control siRNA treatments alone, data presented in [4] were analysed. Error bars show s.e.m. values obtained across experiments or cells as indicated in figure legends. $p$-values representing significance were obtained using the Mann–Whitney $U$-test, proportion test, or paired sample $t$-test or as stated in the corresponding figure legends.

Data accessibility. Additional data are available in the electronic supplementary material, which includes three supplementary movie files.

Authors' contributions. M.H., R.L.S. and I.K. designed and performed experiments and analysed and interpreted data. M.H., R.L.S., I.Z., D.C., P.G. and I.K. analysed data, and prepared figure subpanels. D.D. supported statistical data analysis and DHC-GFP imaging. V.M.D. planned the study, discussed the experimental design, analysed and interpreted data, and wrote the manuscript. V.M.D., M.H. and D.D. edited the manuscript.

Competing interests. The authors declare no competing financial interests.

Funding. This work was supported by a Cancer Research UK Career Development Award (C28598/A9787), a Biotechnology and Biological Sciences Research Council Project grant (BB/R01003X/1), and a Queen Mary, University of London Laboratory start-up grant to V.M.D., a Queen Mary, University of London PhD studentship to M.H., a Biotechnology and Biological Sciences Research Council LIDO-DTP studentship to D.D., an Islamic Bank Development (IDB) PhD studentship to P.G. and a Universiti Brunei Darussalam PhD studentship to I.Z.

Acknowledgements. We thank Jorn Breumlund (GE Healthcare) for technical support while performing TIRF microscopy using OMX-SR™, and the Pines group (ICR) for the plasmid vector-encoding pericentrin fragment.

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
