## [Reviewer comments · Open Biology]

Review History

RSOB-18-0263.R0 (Original submission)

Review form: Reviewer 1

Recommendation

Major revision is needed (please make suggestions in comments)

Are each of the following suitable for general readers?

- a) **Title**
Yes

- b) **Summary**
Yes

- c) **Introduction**
Yes

Is the length of the paper justified?

Yes

Should the paper be seen by a specialist statistical reviewer?

No

Is it clear how to make all supporting data available?

Yes

Is the supplementary material necessary; and if so is it adequate and clear?

Yes

Do you have any ethical concerns with this paper?

No

Comments to the Author

MARK2, also known as Par1b, is a protein kinase that contributes to regulation of cell polarity in response to signals from microtubules and actin. This manuscript describes a new activity-dependent localization for MARK2 at actin-rich retraction fibres. These allow cells to retain attachment to the substratum during mitosis and provide a memory of interphase cytoskeleton organization that dictates spindle orientation. Functional studies undertaken through MARK2 depletion also suggest new roles in promoting spindle positioning following disturbance of the actin network, and in G1/S progression.

While the manuscript is very well written, I'm concerned that the main conclusions from this study are not well supported by the data presented. The inclusion of appropriate controls would also strengthen confidence in the study. Finally, the final result on cell cycle progression feels rather isolated and does not fit well with or add to the main story. In my view, major revisions as detailed below are therefore required before this manuscript can be considered ready for publication in Open Biology.

Figure 1. YFP-MARK2 co-localises as expected to the cortical membrane in interphase cells. However, it is suggested that the non-cortical protein assembles into either vesicles (wild-type and T595E) or striations (kinase-dead) that align along actin filaments. However, stronger actin staining, higher magnifications and some of quantification of the images in panel E are required to confirm this stated correlation of MARK2 and actin distribution in these cells. Also co-staining with a vesicular marker is required to conclude that the 'foci' in panels D and E are vesicles.

Figure 2. The first major conclusion of this study is that MARK2 localizes to retraction fibres in mitotic cells. However, as it stands, the images are very weak and the imaging data needs significant improvement with much larger magnifications to draw this conclusion. Moreover, the authors need to explain more carefully what they are presenting in panel E as the text states that there is no significant difference between the wild-type and mutant proteins, but the histograms show a clear difference. Furthermore, this experiment uses the kinesin inhibitor, STLC, to arrest cells in mitosis, which in itself could alter MARK2 localization. The experiment should therefore be repeated using an alternative approach to arrest cells in mitosis, e.g. MG132.

Figure 3. The data presented in this figure essentially report that LGN and microtubules are not required for localization of MARK2 to the mitotic cortex. Control data under the experimental conditions used here to show loss of LGN by Western blot and the microtubule network by IF should be included rather than simply referencing other work.

Figure 4. Although magnifications are included, the imaging data presented in this figure is very small and difficult to see. Hence, as it stands, it is not convincing of localization of MARK2 to retraction fibres in the presence and absence of actin. Much larger, brighter images should be provided so that MARK2 and actin staining are obvious. A Western blot should also be presented, at least in Supplementary Figures, of the depletion of MARK2.

Figure 5. This figure presents the second major conclusion but relies on a single MARK2 siRNA. Besides repeating with the second siRNA, the authors should present a rescue experiment with the wild-type and kinase-dead MARK2 to know whether this phenotype is dependent on MARK2 activity. The authors should also present the same timings for control versus MARK2 siRNA in panel A so that the delay can be properly assessed, while panels B and C would be more easily interpreted if they were presented as time-courses of % cells that had achieved spindle centering (in addition to the % that had achieved centering within an 8 minute time window).

Figure 6. This result has no apparent relation to the role of MARK2 in spindle positioning and would be better omitted to maintain coherence of the story. If retained, then a flow cytometry time course should be undertaken with at least two independent siRNAs to demonstrate the shift from cell cycle progression to G1/S arrest.

Minor points

- The introductory sentence in the Results on p4 is not helpful as it mentions MARK2 localization in polarized cells. Yet the study uses HeLa cells that do not polarize.
- The statement on p4 that “phosphorylation of MARK2 at T595 by aPKC is insufficient to displace MARK from the plasma membrane” is overstated as it relies on observation of a phosphomimetic mutant; these mutants do not always reliably reflect the properties of the phosphorylated protein.
- Fig. 1A includes mention of STLC when this isn't used in this experiment.
- Fig. 1B should include a Western blot with MARK2 antibodies to demonstrate the relative expression of endogenous and ectopic proteins.
- The orange arrows in panel 1E do not obviously show vesicles “that are arranged parallel to actin stress fibres” as stated in the legend. Again higher magnification of images is required to make this point.
- The authors need to define how they score retraction fibres as short or long in Fig. 2E.
- The text on p5 currently omits any mention of Figs. 3A-D.
- Co-localization data with RFP-PACT showing that YFP-MARK2 is present at the centrosome is shown in Supplementary Figure 2A. I suggest that this is moved to the main Figure 2.

Review form: Reviewer 2

Recommendation

Major revision is needed (please make suggestions in comments)

Are each of the following suitable for general readers?

- a) **Title**
Yes
- b) **Summary**
Yes

c) **Introduction**
Yes

Is the length of the paper justified?

Yes

Should the paper be seen by a specialist statistical reviewer?

Yes

Is it clear how to make all supporting data available?

Yes

Is the supplementary material necessary; and if so is it adequate and clear?

Yes

Do you have any ethical concerns with this paper?

No

Comments to the Author

The correct orientation of the mitotic spindle is the key for accurate cell division and epithelial morphogenesis. In most instances, precise spindle positioning in mammalian cells requires either the interaction of astral microtubules with the cortical dynein and/or robust subcortical actin cytoskeleton. In this manuscript, the authors are extending their previous work on the role of MARK2 in proper spindle orientation, and also revealing its function in postmitotic cells. They show that in addition to its localization at the centrosomes and cell cortex as previously reported, is also localizes to actin-rich adhesion fibers and this localization depends on its kinase activity. They further sought out to reveal the relationship between the actin cytoskeleton and MARK2 in re-centering the spindle during metaphase.

This manuscript by Hart et al., is interesting as the authors have attempted to characterize further the role of MARK2 on various aspects of cell division. However, some part of the current manuscript is unfocused, and lacking several important controls (see below), and I think such issues should be addressed before publication.

Major points:

1. The authors have attempted to address the function of Wild-type, kinase dead (MARK2 D157A) and aPKC phosphomimetic mutant of MARK2 (MARK2 T595E) in the presence of the endogenous proteins that may make interpretation of the results difficult. Does such mutants interact with endogenous MARK2? If yes, all the key experiments must be performed in the absence of endogenous protein. Since the authors have access to the MARK2 antibodies (published by them earlier), it would be critical to see the amount of doxycycline induced exogenous protein with respect to the endogenous protein, to test if such localization (Figure 1 and 2) is not just an artifact of the excess protein induced upon doxycycline-mediated induction. Similarly, MARK2 (WT, KD or T595E) localization at the centrosome (in the absence and presence of Nocodazole) must be analyzed in the absence of endogenous protein.
2. Though it is clear that dynein pathway may not be involved in MARK2-dependent re-centering of the mitotic spindle as Dynein complex or its associated adaptors (NuMA/LGN) localizes at the polar region of the cell cortex, as also mentioned by the authors in the Discussion (p.9). However, while they still conducted LGN-depletion (Figure 3B) to test the role of cortical

dynein for MARK2 localization, it is required that they show the extent of LGN depletion in their experimental condition. Also I feel that Figure 3 can be moved to Supplemental Figures.

3. I found it surprising that authors did not test the function of MARK2 mutants on spindle centering? Since these mutant proteins reveal differential localization pattern of MARK2 (Figure IE), what is the impact of such localization on spindle orientation would be important to know.
4. Also, for Figure 5C (MARK2 siRNA+LAT), the number of cells are too low (n=6) in contrast to control (LAT) where n=51 to conclude 'that MARK2 can correct spindle off-centering induced by actin perturbation'. This experiment surely must be done with large numbers of cells to make such a conclusion. Further, I don't understand why (MARK2 siRNA+LAT) cells are showing pronounced chromosomes instability (Figure 5A), and therefore, it should be discussed if authors observed this phenotype in the majority of the cells.
5. In their last experimental figure, the authors made a sudden jump to study the role of MARK2 in interphase as they observed a reduction in the number of mitotic cells upon MARK2 depletion. They found a decrease in the levels of cyclin B1; I wonder if this happens at the protein level or the transcript level? How authors envisage that MARK2 that localizes at the cell cortex, centrosomes controls cyclin B? what is the levels of MARK2 in different phases of the cell cycle? Do ectopic expression of MARK2 rescue the impact of its depletion on cell cycle? All such remaining questions must be addressed, or at least should be discussed.

Decision letter (RSOB-18-0263.R0)

29-Jan-2019

Dear Dr Draviam,

We are writing to inform you that the Editor has reached a decision on your manuscript RSOB-18-0263 entitled "MARK2/Par1b present at retraction fibres corrects spindle off-centering induced by actin disassembly", submitted to Open Biology.

As you will see from the reviewers' comments below, there are a number of criticisms that prevent us from accepting your manuscript at this stage. The reviewers suggest, however, that a revised version could be acceptable, if you are able to address their concerns. If you think that you can deal satisfactorily with the reviewer's suggestions, we would be pleased to consider a revised manuscript.

The revision will be re-reviewed, where possible, by the original referees. As such, please submit the revised version of your manuscript within six weeks. If you do not think you will be able to meet this date please let us know immediately.

When submitting your revised manuscript, please respond to the comments made by the referee(s) and upload a file "Response to Referees" in "Section 6 - File Upload". You can use this to document any changes you make to the original manuscript. In order to expedite the processing of the revised manuscript, please be as specific as possible in your response to the referee(s).

Please see our detailed instructions for revision requirements
<https://royalsociety.org/journals/authors/author-guidelines/>

Sincerely,

The Open Biology Team
mailto: openbiology@royalsociety.org

Reviewer(s)' Comments to Author(s):

Referee: 1

Comments to the Author(s)

MARK2, also known as Par1b, is a protein kinase that contributes to regulation of cell polarity in response to signals from microtubules and actin. This manuscript describes a new activity-dependent localization for MARK2 at actin-rich retraction fibres. These allow cells to retain attachment to the substratum during mitosis and provide a memory of interphase cytoskeleton organization that dictates spindle orientation. Functional studies undertaken through MARK2 depletion also suggest new roles in promoting spindle positioning following disturbance of the actin network, and in G1/S progression.

While the manuscript is very well written, I'm concerned that the main conclusions from this study are not well supported by the data presented. The inclusion of appropriate controls would also strengthen confidence in the study. Finally, the final result on cell cycle progression feels rather isolated and does not fit well with or add to the main story. In my view, major revisions as detailed below are therefore required before this manuscript can be considered ready for publication in Open Biology.

Figure 1. YFP-MARK2 co-localises as expected to the cortical membrane in interphase cells. However, it is suggested that the non-cortical protein assembles into either vesicles (wild-type and T595E) or striations (kinase-dead) that align along actin filaments. However, stronger actin staining, higher magnifications and some of quantification of the images in panel E are required to confirm this stated correlation of MARK2 and actin distribution in these cells. Also co-staining with a vesicular marker is required to conclude that the 'foci' in panels D and E are vesicles.

Figure 2. The first major conclusion of this study is that MARK2 localizes to retraction fibres in mitotic cells. However, as it stands, the images are very weak and the imaging data needs significant improvement with much larger magnifications to draw this conclusion. Moreover, the authors need to explain more carefully what they are presenting in panel E as the text states that there is no significant difference between the wild-type and mutant proteins, but the histograms show a clear difference. Furthermore, this experiment uses the kinesin inhibitor, STLC, to arrest

cells in mitosis, which in itself could alter MARK2 localization. The experiment should therefore be repeated using an alternative approach to arrest cells in mitosis, e.g. MG132.

Figure 3. The data presented in this figure essentially report that LGN and microtubules are not required for localization of MARK2 to the mitotic cortex. Control data under the experimental conditions used here to show loss of LGN by Western blot and the microtubule network by IF should be included rather than simply referencing other work.

Figure 4. Although magnifications are included, the imaging data presented in this figure is very small and difficult to see. Hence, as it stands, it is not convincing of localization of MARK2 to retraction fibres in the presence and absence of actin. Much larger, brighter images should be provided so that MARK2 and actin staining are obvious. A Western blot should also be presented, at least in Supplementary Figures, of the depletion of MARK2.

Figure 5. This figure presents the second major conclusion but relies on a single MARK2 siRNA. Besides repeating with the second siRNA, the authors should present a rescue experiment with the wild-type and kinase-dead MARK2 to know whether this phenotype is dependent on MARK2 activity. The authors should also present the same timings for control versus MARK2 siRNA in panel A so that the delay can be properly assessed, while panels B and C would be more easily interpreted if they were presented as time-courses of % cells that had achieved spindle centering (in addition to the % that had achieved centering within an 8 minute time window).

Figure 6. This result has no apparent relation to the role of MARK2 in spindle positioning and would be better omitted to maintain coherence of the story. If retained, then a flow cytometry time course should be undertaken with at least two independent siRNAs to demonstrate the shift from cell cycle progression to G1/S arrest.

Minor points

- The introductory sentence in the Results on p4 is not helpful as it mentions MARK2 localization in polarized cells. Yet the study uses HeLa cells that do not polarize.
- The statement on p4 that “phosphorylation of MARK2 at T595 by aPKC is insufficient to displace MARK from the plasma membrane” is overstated as it relies on observation of a phosphomimetic mutant; these mutants do not always reliably reflect the properties of the phosphorylated protein.
- Fig. 1A includes mention of STLC when this isn't used in this experiment.
- Fig. 1B should include a Western blot with MARK2 antibodies to demonstrate the relative expression of endogenous and ectopic proteins.
- The orange arrows in panel 1E do not obviously show vesicles “that are arranged parallel to actin stress fibres” as stated in the legend. Again higher magnification of images is required to make this point.
- The authors need to define how they score retraction fibres as short or long in Fig. 2E.
- The text on p5 currently omits any mention of Figs. 3A-D.
- Co-localization data with RFP-PACT showing that YFP-MARK2 is present at the centrosome is shown in Supplementary Figure 2A. I suggest that this is moved to the main Figure 2.

Referee: 2

Comments to the Author(s)

The correct orientation of the mitotic spindle is the key for accurate cell division and epithelial morphogenesis. In most instances, precise spindle positioning in mammalian cells requires either the interaction of astral microtubules with the cortical dynein and/or robust subcortical actin

cytoskeleton. In this manuscript, the authors are extending their previous work on the role of MARK2 in proper spindle orientation, and also revealing its function in postmitotic cells. They show that in addition to its localization at the centrosomes and cell cortex as previously reported, it also localizes to actin-rich adhesion fibers and this localization depends on its kinase activity. They further sought out to reveal the relationship between the actin cytoskeleton and MARK2 in re-centering the spindle during metaphase.

This manuscript by Hart et al., is interesting as the authors have attempted to characterize further the role of MARK2 on various aspects of cell division. However, some part of the current manuscript is unfocused, and lacking several important controls (see below), and I think such issues should be addressed before publication.

Major points:

1. The authors have attempted to address the function of Wild-type, kinase dead (MARK2 D157A) and aPKC phosphomimetic mutant of MARK2 (MARK2 T595E) in the presence of the endogenous proteins that may make interpretation of the results difficult. Does such mutants interact with endogenous MARK2? If yes, all the key experiments must be performed in the absence of endogenous protein. Since the authors have access to the MARK2 antibodies (published by them earlier), it would be critical to see the amount of doxycycline induced exogenous protein with respect to the endogenous protein, to test if such localization (Figure 1 and 2) is not just an artifact of the excess protein induced upon doxycycline-mediated induction. Similarly, MARK2 (WT, KD or T595E) localization at the centrosome (in the absence and presence of Nocodazole) must be analyzed in the absence of endogenous protein.
2. Though it is clear that dynein pathway may not be involved in MARK2-dependent re-centering of the mitotic spindle as Dynein complex or its associated adaptors (NuMA/LGN) localizes at the polar region of the cell cortex, as also mentioned by the authors in the Discussion (p.9). However, while they still conducted LGN-depletion (Figure 3B) to test the role of cortical dynein for MARK2 localization, it is required that they show the extent of LGN depletion in their experimental condition. Also I feel that Figure 3 can be moved to Supplemental Figures.
3. I found it surprising that authors did not test the function of MARK2 mutants on spindle centering? Since these mutant proteins reveal differential localization pattern of MARK2 (Figure IE), what is the impact of such localization on spindle orientation would be important to know.
4. Also, for Figure 5C (MARK2 siRNA+LAT), the number of cells are too low (n=6) in contrast to control (LAT) where n=51 to conclude 'that MARK2 can correct spindle off-centering induced by actin perturbation'. This experiment surely must be done with large numbers of cells to make such a conclusion. Further, I don't understand why (MARK2 siRNA+LAT) cells are showing pronounced chromosomes instability (Figure 5A), and therefore, it should be discussed if authors observed this phenotype in the majority of the cells.
5. In their last experimental figure, the authors made a sudden jump to study the role of MARK2 in interphase as they observed a reduction in the number of mitotic cells upon MARK2 depletion. They found a decrease in the levels of cyclin B1; I wonder if this happens at the protein level or the transcript level? How authors envisage that MARK2 that localizes at the cell cortex, centrosomes controls cyclin B? what is the levels of MARK2 in different phases of the cell cycle? Do ectopic expression of MARK2 rescue the impact of its depletion on cell cycle? All such remaining questions must be addressed, or at least should be discussed.

Author's Response to Decision Letter for (RSOB-18-0263.R0)

See Appendix A.

RSOB-18-0263.R1 (Revision)

Review form: Reviewer 2

Recommendation

Accept with minor revision (please list in comments)

Are each of the following suitable for general readers?

- a) **Title**
Yes
- b) **Summary**
Yes
- c) **Introduction**
Yes

Is the length of the paper justified?

Yes

Should the paper be seen by a specialist statistical reviewer?

No

Is it clear how to make all supporting data available?

Yes

Is the supplementary material necessary; and if so is it adequate and clear?

Yes

Do you have any ethical concerns with this paper?

No

Comments to the Author

Authors have substantially improved the revised version of this manuscript and have sufficiently addressed my previous concerns.

I have now only a few minor concerns:

1. In the Supplemental Figure 1D, immunoblot related to the expression of KD in the absence of endogenous is not convincing. Expression of MARK2 in control is anyway weak, and in cells lacking endogenous MARK2 by siRNAs, I could still observe a faint band. Also, Authors should provide molecular weight (in kDa) for this immunoblot.

2. Also for Figure S1, I feel that in the absence of endogenous protein, the cell expressing MARK2-YFP-KD barely show prominent long striations in contrast to the cell having endogenous protein (compare Figure 1E middle panel with the Supplemental Figure S1C bottom panel). I suggest that authors quantify this phenotype, and if they feel that the striations are reduced upon removal of endogenous protein, they must report this. Also, it appears to me the KD cell in S1C is more spread out than the control cell, and thus, I would request authors to show phenotypically similar cells for such a phenotype.

Decision letter (RSOB-18-0263.R1)

20-May-2019

Dear Dr Draviam,

We are pleased to inform you that your manuscript RSOB-18-0263.R1 entitled "MARK2/Par1b present at retraction fibres corrects spindle off-centering induced by actin disassembly" has been accepted by the Editor for publication in Open Biology. The reviewer(s) have recommended publication, but also suggest some minor revisions to your manuscript. Therefore, we invite you to respond to the reviewer(s)' comments and revise your manuscript.

Please submit the revised version of your manuscript within 7 days. If you do not think you will be able to meet this date please let us know immediately and we can extend this deadline for you.

- 1) A text file of the manuscript (doc, txt, rtf or tex), including the references, tables (including captions) and figure captions. Please remove any tracked changes from the text before submission. PDF files are not an accepted format for the "Main Document".
- 2) A separate electronic file of each figure (tiff, EPS or print-quality PDF preferred). The format should be produced directly from original creation package, or original software format. Please note that PowerPoint files are not accepted.
- 3) Electronic supplementary material: this should be contained in a separate file from the main

text and meet our ESM criteria (see <http://royalsocietypublishing.org/instructions-authors#question5>). All supplementary materials accompanying an accepted article will be treated as in their final form. They will be published alongside the paper on the journal website and posted on the online figshare repository. Files on figshare will be made available approximately one week before the accompanying article so that the supplementary material can be attributed a unique DOI.

Online supplementary material will also carry the title and description provided during submission, so please ensure these are accurate and informative. Note that the Royal Society will not edit or typeset supplementary material and it will be hosted as provided. Please ensure that the supplementary material includes the paper details (authors, title, journal name, article DOI). Your article DOI will be 10.1098/rsob.2016[last 4 digits of e.g. 10.1098/rsob.20160049].

4) A media summary: a short non-technical summary (up to 100 words) of the key findings/importance of your manuscript. Please try to write in simple English, avoid jargon, explain the importance of the topic, outline the main implications and describe why this topic is newsworthy.

Images

Data-Sharing

It is a condition of publication that data supporting your paper are made available. Data should be made available either in the electronic supplementary material or through an appropriate repository. Details of how to access data should be included in your paper. Please see <http://royalsocietypublishing.org/site/authors/policy.xhtml#question6> for more details.

Data accessibility section

Sincerely,

The Open Biology Team

<mailto:openbiology@royalsociety.org>

Reviewer(s)' Comments to Author:

Referee: 2

Comments to the Author(s)

Authors have substantially improved the revised version of this manuscript and have sufficiently addressed my previous concerns.

I have now only a few minor concerns:

1. In the Supplemental Figure 1D, immunoblot related to the expression of KD in the absence of endogenous is not convincing. Expression of MARK2 in control is anyway weak, and in cells lacking endogenous MARK2 by siRNAs, I could still observe a faint band. Also, Authors should provide molecular weight (in kDa) for this immunoblot.
2. Also for Figure S1, I feel that in the absence of endogenous protein, the cell expressing MARK2-YFP-KD barely show prominent long striations in contrast to the cell having endogenous protein (compare Figure 1E middle panel with the Supplemental Figure S1C bottom panel). I suggest that authors quantify this phenotype, and if they feel that the striations are reduced upon removal of endogenous protein, they must report this. Also, it appears to me the KD cell in S1C is more spread out than the control cell, and thus, I would request authors to show phenotypically similar cells for such a phenotype.

Author's Response to Decision Letter for (RSOB-18-0263.R1)

See Appendix B.

Decision letter (RSOB-18-0263.R2)

30-May-2019

Dear Dr Draviam

We are pleased to inform you that your manuscript entitled "MARK2/Par1b kinase present at centrosomes and retraction fibres corrects spindle off-centering induced by actin disassembly" has been accepted by the Editor for publication in Open Biology.

Article processing charge

Please note that the article processing charge is immediately payable. A separate email will be sent out shortly to confirm the charge due. The preferred payment method is by credit card; however, other payment options are available.

Sincerely,

The Open Biology Team
mailto: openbiology@royalsociety.org

Response to Referee comments

Referee 1

We thank the reviewer for the highly constructive comments. The reviewer acknowledges that the “the manuscript is very well written” and has added that “appropriate controls would also strengthen confidence in the study.” We have addressed all the queries (details listed below) and include several pieces of control data to strengthen the main findings. We have removed the cell cycle data that both reviewers have commented as feeling isolated from the main story.

1. “... stronger actin staining, higher magnifications and some of quantification of the images in panel E are required to confirm this stated correlation of MARK2 and actin distribution in these cells. Also co-staining with a vesicular marker is required to conclude that the ‘foci’ in panels D and E are vesicles”.

As suggested, we provide stronger actin staining using images acquired with longer exposure times and also we include magnified insets (Figure 1E). We agree that we do not define the precise vesicular compartment(s) occupied. So we have changed ‘vesicular’ signal into YFP ‘foci’ throughout the manuscript for accuracy. We thank the referee for highlighting this weakness.

As recommended, we provide line profiles (Figure 1F) to represent quantification of YFP intensities shown in Figure 1E. We have also incorporated number of cells that correspond to each phenotype.

2. “the images are very weak and the imaging data needs significant improvement with much larger magnifications to draw this conclusion. Moreover, the authors need to explain more carefully what they are presenting in panel E as the text states that there is no significant difference between the wild-type and mutant proteins, but the histograms show a clear difference. Furthermore, this experiment uses the kinesin inhibitor, STLC, to arrest cells in mitosis, which in itself could alter MARK2 localization. The experiment should therefore be repeated using an alternative approach to arrest cells in mitosis, e.g. MG132”

2a. As suggested, we include larger magnifications of images and images that exclude the use of any drug including the kinesin inhibitor STLC (Supplementary Figure-2).

2b. We clarify panel E text in Page-5; 2nd paragraph.

2c. About STLC use: In YFP-MARK2 kinase dead expressing cells, bipolar spindles were frequently tumbled. To exclude the possibility of interference from these tumbled spindles, we treated cells with STLC (an Eg5 inhibitor) to create monopolar spindles that are uniformly positioned in WT and KD expressing cells - we now include this reasoning in the results text (Page 5, first paragraph). To quantify MARK2 signals in cells with comparable spindle position is important and so we have retained STLC treated cells in the Main figure and included the new images of drug untreated controls recommended by the reviewer as Supplementary figure-2 - the new images confirm results from previous imaging from monopolar spindles.

3. “Control data under the experimental conditions used here to show loss of LGN by Western blot and the microtubule network by IF should be included rather than simply referencing other work.”

As suggested, we now include a blot showing the extent of LGN depletion (Supplementary Figure 4B) and immunofluorescence (IF) images showing the loss of microtubules in Nocodazole treated cells (Figure 3G).

4. “the imaging data presented in this figure is very small and difficult to see. Hence, as it stands, it is not convincing of localization of MARK2 to retraction fibres in the presence and absence of actin. Much larger, brighter images should be provided so that MARK2 and actin staining are obvious. A Western blot should also be presented, at least in Supplementary Figures, of the depletion of MARK2.”

We have now include larger images (30% larger than before). To allow this enlargement in size, we have split Figure-4 into two Figures 4 and 5. We have also included a western-blot to indicate the extent of MARK2 depletion as recommended (Fig 5A).

5a. “Besides repeating with the second siRNA, the authors should present a rescue experiment with the wild-type and kinase-dead MARK2 to know whether this phenotype is dependent on MARK2 activity.”

Latrunculin treated cells are photo-sensitive and they frequently arrest or die prior to anaphase. So, we have had to run multiple repeats of imaging to increase the number of MARK2-depleted mitotic cells that completed metaphase-anaphase transition successfully - we have now nearly tripled the sample size and the outcomes are similar to what we reported previously based on smaller sample size (Fig. 6). The second siRNA oligo for MARK2 unfortunately yielded severe mitotic cell death in the presence of Latrunculin rendering the movies useless for quantitative analysis. We faced this difficulty in addition to the reduced number of mitotic cells observed following MARK2 depletion (Previous Figure-6 on cell cycle impact; now omitted).

We are unable to perform rescue experiments because the HeLa FRT/TO cells are very small to detect equatorial off-centering (see Reviewer 2 comment 3). However we have been able to quantitatively compare pre-anaphase and anaphase movements (Sup. Fig. 5 and Fig. 6) and show “MARK2’s correction role to be specifically relevant for pre-anaphase stage of mitosis.”

5b. “The authors should also present the same timings for control versus MARK2 siRNA in panel A so that the delay can be properly assessed, Although the same timings can not be presented because different cells (within the same treatment condition) can display different lengths of time in prometaphase and metaphase, we present Supplementary movies that include all time points. In the figure, panel A, we have included the times that highlight spindle off-centering phenotype *per se*.”

5c. “while panels B and C would be more easily interpreted if they were presented as time-courses of % cells that had achieved spindle centering (in addition to the % that had achieved centering within an 8 minute time window).”

We clarify that unlike Latrunculin untreated cells, Latrunculin treated cells do not achieve/maintain stable spindle centering. Hence, we can not describe the % of cells that had achieved spindle centering in the presence of Latrunculin-A. However, Latrunculin-A treatment allows us to compare the Off-centering to Centering (OC to C) conversion rates in Control *versus* MARK2 siRNA treated cells, which is described in panels B and C and Sup Fig 5. We clarify this in the text in Page-8.

6. “This result has no apparent relation to the role of MARK2 in spindle positioning and would be better omitted to maintain coherence of the story.”

As both reviewer’s find that this piece of data appears to be unrelated to the main manuscript we have removed this result.

Minor points

(i) “The introductory sentence in the Results on p4 is not helpful as it mentions MARK2 localization in polarized cells. Yet the study uses HeLa cells that do not polarize.”

Included.

(ii) “The statement on p4 that “phosphorylation of MARK2 at T595 by aPKC is insufficient to displace MARK from the plasma membrane” is overstated as it relies on observation of a phosphomimetic mutant”

We agree and have reworded “the phosphorylation of MARK2 at T595 by aPKC **may be** insufficient to displace MARK2 from the plasma membrane.”

(iii) “Fig. 1A includes mention of STLC when this isn’t used in this experiment”

Removed.

(iv) “Fig. 1B should include a Western blot with MARK2 antibodies to demonstrate the relative expression of endogenous and ectopic proteins”.

Included (right image; Fig 1B).

(v) “The orange arrows in panel 1E do not obviously show vesicles “that are arranged parallel to actin stress fibres” as stated in the legend. Again higher magnification of images is required to make this point.”

New images included in panel 1E along with a quantification panel 1F.

(vi) “The authors need to define how they score retraction fibres as short or long in Fig. 2E.”

Described in Figure 2 legend and text.

(vii) “The text on p5 currently omits any mention of Figs. 3A-D.”

Apologies for the omission. Now Included.

(viii) “Co-localization data with RFP-PACT showing that YFP-MARK2 is present at the centrosome is shown in Supplementary Figure 2A. I suggest that this is moved to the main Figure 2.”

We have separated centrosome data from Figure -2 to give maximum focus on retraction fibres (the main purpose of the paper). Centrosome localisation is retained in Supplementary figure 3. We are happy to include it as a separate main figure, if Referee-1 feels it must be part of Main figures.

Referee 2

We thank the referee for highly constructive comments and suggestions on controls required. We have addressed these through new experiments, quantitative analysis and text changes as listed below.

1a. "... endogenous proteins that may make interpretation of the results difficult. Does such mutants interact with endogenous MARK2? If yes, all the key experiments must be performed in the absence of endogenous protein."

This is a valid concern but difficult to address for a kinase. We have redone most key experiments in the absence of endogenous MARK2 where possible (listed below). We do not emphasise this type of 'endogenous protein depletion' experiment in this paper, although we have routinely used this method in other protein depletions published from the group (Shrestha et al. 2014; Shrestha et al. 2017) because there is no easy way to assess the extent of endogenous MARK2 depletion at the single-cell level. So, we include a discussion paragraph the possibility of rescue through low-levels of non-depleted endogenous MARK2, to make sure that the readers are aware of the limitations of our interpretations. I hope the new experiments and rewording of text will address this reviewer' comment satisfactorily.

List of new experiments involving endogenous protein depletion:

1. Figure 1B: We include an immunoblot to show the conditions for the depletion endogenous MARK2 and conditional expression of low levels of siRNA-resistant MARK2 variants in HeLa cells.
2. Page 4: Analysis of YFP-MARK2 signals in WT or KD expressing cells in interphase, depleted of endogenous MARK2, showed densely arranged YFP foci in WT cells and long striations of YFP in addition to the foci in KD expressing cells (Supplementary Fig 1C) with a corresponding immunoblot (Supplementary Fig 1B).
3. Page 5: Analysis of YFP-MARK2 signals at the retraction fibres of WT or KD expressing cells, depleted of endogenous MARK2, showed reduced YFP localisation at retraction fibres mitotic imaging (Supplementary Fig 2B)
4. Page 6: We assess the presence of YFP-MARK2 (WT and KD) at spindle poles in endogenous MARK2 depleted cells (Supplementary Fig 3C).

1b. "Since the authors have access to the MARK2 antibodies (published by them earlier), it would be critical to see the amount of doxycycline induced exogenous protein with respect to the endogenous protein, to test if such localization (Figure 1 and 2) is not just an artifact of the excess protein induced upon doxycycline-mediated induction "

To show levels of endogenous MARK2 and compare it across the various siRNA resistant MARK2 variants, we include immunoblots that show no dramatic increase in MARK2 expression following Doxycycline treatment (Figure 1B).

1c. "Similarly, MARK2 (WT, KD or T595E) localization at the centrosome (in the absence and presence of Nocodazole) must be analyzed in the absence of endogenous protein."

We have tested the localisation of MARK2 variants at the centrosome in the absence of endogenous protein to conclude that the centrosome localisation is not due to differences in levels of expression (Sup Fig 3C). However, we are aware that there is a remote possibility

of the endogenous MARK2 enabling the centrosomal localisation of the MARK2-WT protein. So, we discuss the inability to completely exclude the contribution of endogenous MARK2, and highlight the need for future deletion studies to identify the precise regulation of MARK2 localisation at centrosomes (Page 9, para-2 (Discussion)).

2. “it is required that they show the extent of LGN depletion in their experimental condition. Also I feel that Figure 3 can be moved to Supplemental Figures.”

We move LGN studies (including LGN depletion extent) to Supplementary figure-4 as this is supportive evidence for Figure 4 where we show that LGN and cortical Dynein occupy distinct areas. However, the rest of figure-3 is very important because it excludes a feedback regulation of MARK2 localisation by cortex-microtubule interaction (See below). We have shown that the depletion of MARK2 leads to longer astral microtubules (Zulkipli et al., JCB 2019); whether microtubule length in turn has an impact on MARK2 localisation - as negative feedback loop- is not known. This is possible because in *C.elegans* MARK2/PAR1 levels at the cortex are dynamically regulated, and so it's important to study factors that can influence MARK2's cortical localisation in human cells. We now clarify the above purpose and significance of Figure-3 in the results section (Page 6) and last paragraph of discussion.

3. “I found it surprising that authors did not test the function of MARK2 mutants on spindle centering? Since these mutant proteins reveal differential localization pattern of MARK2 (Figure IE), what is the impact of such localization on spindle orientation would be important to know.”

We could not perform these studies fully because of technical reasons. We spent over a year to generate HeLa-FRT/TO cell lines expressing MARK2-WT and -KD with great enthusiasm and effort but unfortunately these cells are ~20% smaller than HeLa cells and in these cells OC measurements are nearly impossible - this is the primary reason for not being able to test. In addition MARK2-KD expressing cells present increased occurrence of tumbled spindles (a phenotype not observed following MARK2 depletion), possibly indicating a dominant negative function for the KD mutant. Nevertheless, the cell lines will be useful for the field for biochemical or cell biological studies and so we present them here to explain the factors that control MARK2 localisation.

4a. “... for Figure 5C (MARK2 siRNA+LAT), the number of cells are too low (n=6) in contrast to control (LAT) where n=51 to conclude ‘that MARK2 can correct spindle off-centering induced by actin perturbation’. This experiment surely must be done with large numbers of cells to make such a conclusion.”

As mentioned in the manuscript (Page-7) in MARK2 siRNA treated cells exposed to Latrunculin “the number of cells that initiated mitosis was reduced”. To increase sample size, we have rerun the study several times and have tripled the cell numbers - these are consistent with our original data that MARK2 is required to correct spindle off-centering induced by actin perturbation (Figure 6C).

4b. “...why (MARK2 siRNA+LAT) cells are showing pronounced chromosomes instability (Figure 5A), and therefore, it should be discussed if authors observed this phenotype in the majority of the cells.”

We thank the referee for highlighting this point. We now include in page-7 Results text that “a mild congression defect was observed in ~15% of cells” depleted of MARK2 and exposed to Latrunculin.

5. “In their last experimental figure, the authors made a sudden jump to study the role of MARK2 in interphase”

This section has been removed as both referees find this section as a huge jump. There is however no reduction in the number of figures as we have included an additional figure to clarify reviewer’s queries on MARK2’s mitotic role.

Appendix B

Viji M Draviam, PhD

Reader (Associate Professor) in Cell Biology
Director of Industrial Research and Innovation
School of Biological and Chemical Sciences
QMUL London E1 4NS

Email: v.draviam@qmul.ac.uk

23rd of May 2019

To
The Editor
Open Biology

Ref: Hart et al., (Research Article – Revision-2)

Dear Editor,

I am writing to submit our revised manuscript titled “MARK2/Par1b present at centrosomes and retraction fibres corrects spindle off-centering induced by actin disassembly”.

I am delighted to learn that our revised version is satisfactory and reviewer-2 has only requested for a minor revision of Supplementary figure subpanels 1B and 1C, which we fully address in the new revised version (detailed in Response to reviewer’s comments).

Our current manuscript will fit well in the Open Biology special series called “Focus on Centrosome Biology”, as we show that MARK2/Par1b kinase (an evolutionarily conserved polarity kinase) is a centrosomal protein that localises at centrosomes independent of its kinase activity or microtubule status. To highlight the centrosome part of our study, as recommended by reviewer-1 during the first revision cycle, we have now moved Supplementary figure-3 to Main figure-3 (the finding was already highlighted as part of our abstract; we now highlight it further by separating the results paragraph (without any new text) and include the word ‘centrosome’ in the title as well).

Response to reviewer’s comments:

Reviewer-2 indicates that we have ‘substantially improved the revised version of this manuscript and have sufficiently addressed’ his/her concerns.

Reviewer-2 has correctly highlighted the lack of full depletion of MARK2 in the HeLa FRT/TO lines (Figure S1B) – this is a technical difficulty we face with MARK2 siRNA treatment in the highly sensitive HeLa FRT/TO cell line. To ensure that we accurately describe the extent of endogenous protein depletion, we have now measured protein levels (band intensities) using our fluorescent immuno-blots and have explained in text that the studies are conducted following a reduction, and not complete absence, of endogenous MARK2. Nevertheless, we are confident of our conclusion because of the quantitative assessment in single-cells: As recommended by the reviewer, we have quantified the phenotypes in nearly 40 live-cells and have concluded that the kinase-dead mutant’s

localisation is distinct from that of the Wild-type in all cells observed so far in the presence or absence of MARK2 siRNA treatment.

Finally, the comment on the cell shapes being different in Figure S1C has been resolved by including an additional example of cells with varying shapes – we find no correlation between cell shape and kinase localisation; although we see a strong correlation between kinase activity and kinase localisation. Thus, we conclude that MARK2's intrinsic kinase activity is required for MARK2's punctate localisation in interphase.

I hope you will find the minor revisions satisfactory, and the manuscript acceptable for publication.

I look forward to hearing from you.

Sincerely yours,

Viji M. Draviam